# Genome-wide analysis provides genetic evidence that ACE2 influences COVID-19 risk and yields risk scores associated with severe disease

Julie E. Horowitz[1,5], Jack A. Kosmicki[1,5], Amy Damask[1,5], Deepika Sharma[1], Genevieve H. L. Roberts[2], Anne E. Justice[3], Nilanjana Banerjee[1], Marie V. Coignet[2], Ashish Yadav[1], Joseph B. Leader[3], Anthony Marcketta[1], Danny S. Park[2], Rouel Lanche[1], Evan Maxwell[1], Spencer C. Knight[2], Xiaodong Bai[1], Harendra Guturu[2], Dylan Sun[1], Asher Baltzell[2], Fabricio S. P. Kury[1], Joshua D. Backman[1], Ahna R. Girshick[2], Colm O'Dushlaine[1], Shannon R. McCurdy[2], Raghavendran Partha[2], Adam J. Mansfield[1], David A. Turissini[2], Alexander H. Li[1], Miao Zhang[2], Joelle Mbatchou[1], Kyoko Watanabe[1], Lauren Gurski[1], Shane E. McCarthy[1], Hyun M. Kang[1], Lee Dobbyn[1], Eli Stahl[1], Anurag Verma[4], Giorgio Sirugo[4], Regeneron Genetics Center*, Marylyn D. Ritchie[4], Marcus Jones[1], Suganthi Balasubramanian[1], Katherine Siminovitch[1], William J. Salerno[1], Alan R. Shuldiner[1], Daniel J. Rader[4], Tooraj Mirshahi[3], Adam E. Locke[1], Jonathan Marchini[1], John D. Overton[1], David J. Carey[3], Lukas Habegger[1], Michael N. Cantor[1], Kristin A. Rand[2], Eurie L. Hong[2], Jeffrey G. Reid[1], Catherine A. Ball[2], Aris Baras[1,6], Gonçalo R. Abecasis[1,6] and Manuel A. R. Ferreira[1,6 ✉]

Severe acute respiratory syndrome coronavirus 2 (SARS-CoV-2) enters human host cells via angiotensin-converting enzyme 2 (ACE2) and causes coronavirus disease 2019 (COVID-19). Here, through a genome-wide association study, we identify a variant (rs190509934, minor allele frequency 0.2–2%) that downregulates ACE2 expression by 37% ($P = 2.7 \times 10^{-8}$) and reduces the risk of SARS-CoV-2 infection by 40% (odds ratio = 0.60, $P = 4.5 \times 10^{-13}$), providing human genetic evidence that ACE2 expression levels influence COVID-19 risk. We also replicate the associations of six previously reported risk variants, of which four were further associated with worse outcomes in individuals infected with the virus (in/near LZTFL1, MHC, DPP9 and IFNAR2). Lastly, we show that common variants define a risk score that is strongly associated with severe disease among cases and modestly improves the prediction of disease severity relative to demographic and clinical factors alone.

Coronavirus disease 2019 (COVID-19) is caused by infection with severe acute respiratory syndrome coronavirus 2 (SARS-CoV-2), which enters human host cells via angiotensin-converting enzyme 2 (ACE2)[1]. COVID-19 ranges from asymptomatic infection to severe disease, including respiratory failure and death[2–4], and has led to more than 5 million deaths worldwide since December 2019[5]. Reported risk factors for severe COVID-19 include male sex, older age, ethnicity, obesity and cardiovascular and respiratory diseases[6–8], among others. Host genetic factors have also been shown to modulate the risk of infection and disease severity[9–12]. The largest human genetics study performed so far included data from 49,562 individuals infected with SARS-CoV-2 and >1.7 million individuals with no record of infection as controls, and identified 13 independent common risk variants[12], many located in or near immune-related genes, such as IFNAR2 and CXCR6. Genetic studies of rare variation assayed through exome or genome sequencing have

also suggested a role in COVID-19 for genes in the type 1 interferon (IFN) pathway, including TLR7[13–15]. Still, a complete understanding of genetic susceptibility to SARS-CoV-2 infection and progression to severe COVID-19, and the applicability of these findings for risk prediction, are incompletely understood. In this study, we performed a genome-wide association study (GWAS) meta-analysis to identify additional genetic variants associated with COVID-19 since these may help identify new therapies. We also tested the utility of genetic risk scores (GRS) to identify individuals at the highest risk of severe disease, who could be prioritized for vaccination or therapeutic interventions, which globally are in short supply.

## Results

**GWAS of SARS-CoV-2 infection identifies ACE2 association.** We performed GWAS of COVID-19 outcomes across 52,630 individuals with COVID-19 and 704,016 individuals with no record of

[1]Regeneron Genetics Center, Tarrytown, NY, USA. [2]AncestryDNA, Lehi, UT, USA. [3]Geisinger, Danville, PA, USA. [4]Department of Genetics, Perelman School of Medicine, University of Pennsylvania, Philadelphia, PA, USA. [5]These authors contributed equally: Julie E. Horowitz, Jack A. Kosmicki, Amy Damask. [6]These authors jointly supervised this work: Aris Baras, Goncalo R. Abecasis, Manuel A. R. Ferreira. *A list of members and their affiliations appear at the end of the paper. ✉e-mail: manuel.ferreira@regeneron.com

**Table 1 | The seven COVID-19 phenotypes analyzed in this study**

| Broad phenotype category | Phenotype | Description | Group | Sample size with genetic data |
|---|---|---|---|---|
| Risk of infection | COVID-19 positive versus COVID-19 negative or unknown | Risk of infection | Cases | 52,630 |
| | | | Controls | 704,016 |
| | COVID-19 positive versus COVID-19 negative | Risk of infection among individuals tested for SARS-CoV-2 | Cases | 52,630 |
| | | | Controls | 109,605 |
| | COVID-19 positive and not hospitalized versus COVID-19 negative or unknown | Risk of infection that did not require hospitalization | Cases | 45,641 |
| | | | Controls | 704,016 |
| | COVID-19 positive and hospitalized versus COVID-19 negative or unknown | Risk of infection that required hospitalization | Cases | 6,911 |
| | | | Controls | 689,620 |
| | COVID-19 positive and severe versus COVID-19 negative or unknown | Risk of infection with severe outcomes | Cases | 2,184 |
| | | | Controls | 689,620 |
| Risk of severe outcomes in individuals infected with the virus | COVID-19 positive and hospitalized versus COVID-19 positive and not hospitalized | Risk of hospitalization in individuals infected with the virus | Cases | 6,911 |
| | | | Controls | 45,185 |
| | COVID-19 positive and severe versus COVID-19 positive and not hospitalized | Risk of severe disease in individuals infected with the virus | Cases | 2,184 |
| | | | Controls | 45,185 |

SARS-CoV-2 infection aggregated from 4 studies (Geisinger Health System (GHS), Penn Medicine BioBank (PMBB), UK Biobank (UKB) and AncestryDNA; Supplementary Table 1) and 5 continental ancestries. Of the cases with COVID-19, 6,911 (13.1%) were hospitalized and 2,184 (4.1%) had severe disease; hospitalized patients were more likely to be older, of non-European ancestry and to have preexisting cardiovascular and lung disease (Supplementary Table 2). Using these data, we defined five case-control comparisons related to the risk of infection and two others related to disease severity among cases with COVID-19 (Table 1 and Supplementary Table 3). For each comparison, we performed ancestry-specific GWAS in each study using REGENIE (Methods) and then combined the results using a fixed-effects meta-analysis. Genomic inflation factors ($\lambda_{GC}$) for the meta-analyses were <1.05, suggesting no substantial impact of population structure or unmodeled relatedness (Supplementary Table 4). Unless otherwise noted, all association $P$ values reported henceforth are from Firth (disease traits) or linear (quantitative traits) regression tests performed in REGENIE.

Our analysis provides independent support for several risk variants reported in previous GWAS of COVID-19[9–11] (Supplementary Table 5), including those recently reported by the COVID-19 Host Genetics Initiative (HGI)[12], to which we contributed an earlier version of these data (Supplementary Table 6). Details for these replicated loci follow below, but first we looked for new genetic associations that might have been missed by the HGI. Across the seven risk and severity phenotypes, considering both common (minor allele frequency (MAF) > 0.5%, up to 13 million) and rare (MAF < 0.5%, up to 76 million) variants, we observed one previously unreported association at a conservative $P < 8 \times 10^{-11}$ (Bonferroni correction for seven phenotypes × 89 million variants). This association was between a lower risk of SARS-CoV-2 infection (52,630 cases positive for COVID-19 versus 704,016 COVID-19 negative or unknown controls) and rs190509934:C on the X chromosome (MAF = 0.3%, odds ratio (OR) = 0.60, 95% confidence interval (CI) = 0.52–0.69, $P = 4.5 \times 10^{-13}$; Fig. 1). This rare variant is located 60 base pairs (bp) upstream of the *ACE2* gene (Fig. 2a), the primary cell entry receptor for SARS-CoV-2[16].

Given the potential significance of these findings, we studied the association between the *ACE2* variant rs190509934 and COVID-19 outcomes in greater detail. We found that the variant was well imputed (imputation info score > 0.5 for all studies) and that there was no evidence for differences in effect size (heterogeneity test $P > 0.05$) across studies (Fig. 2b) or ancestries (Supplementary Table 7). However, a significantly stronger association with SARS-CoV-2 infection (heterogeneity test $P = 0.009$) was observed in males (OR = 0.49, $P = 7.0 \times 10^{-11}$, explaining 0.085% of the variance in disease liability[17], $h^2$) when compared to females (OR = 0.72, $P = 5 \times 10^{-4}$; $h^2 = 0.017\%$). There were no associations between rs190509934 and 6 clinical risk factors for COVID-19 after multiple test correction (all with $P > 0.05/6 = 0.008$; Supplementary Table 8), suggesting these did not likely confound the analysis. We then investigated the association between rs190509934 and severity among cases with COVID-19 and found that carriers of rs190509934:C had a numerically (but not significantly) lower risk of worse disease outcomes when compared to non-carriers (for example, OR = 0.69, $P = 0.16$ when comparing 6,779 cases hospitalized with COVID-19 versus 44,968 cases not hospitalized with COVID-19; Supplementary Table 9). These results demonstrate that rs190509934 near *ACE2* confers protection against SARS-CoV-2 infection and potentially also modulates disease severity among individuals infected with the virus; since the variant is relatively uncommon, a definitive account of its role in disease severity requires assessing larger numbers of severe cases.

We speculated that the protective rare variant near *ACE2* (rs190509934:C) might regulate *ACE2* expression. This variant was not characterized by the Genotype-Tissue Expression (GTEx) consortium[18] or 51 other gene expression studies we queried (Supplementary Table 10). Thus, to test its association with *ACE2* expression, we analyzed RNA sequencing (RNA-seq) data from liver tissue available in a subset of 2,035 individuals from the GHS study, including 8 heterozygous and 1 hemizygous carrier for rs190509934:C. After adjusting for potential confounders (for example, body mass index (BMI), liver disease), we found that rs190509934:C reduced *ACE2* expression by 0.87 s.d. units

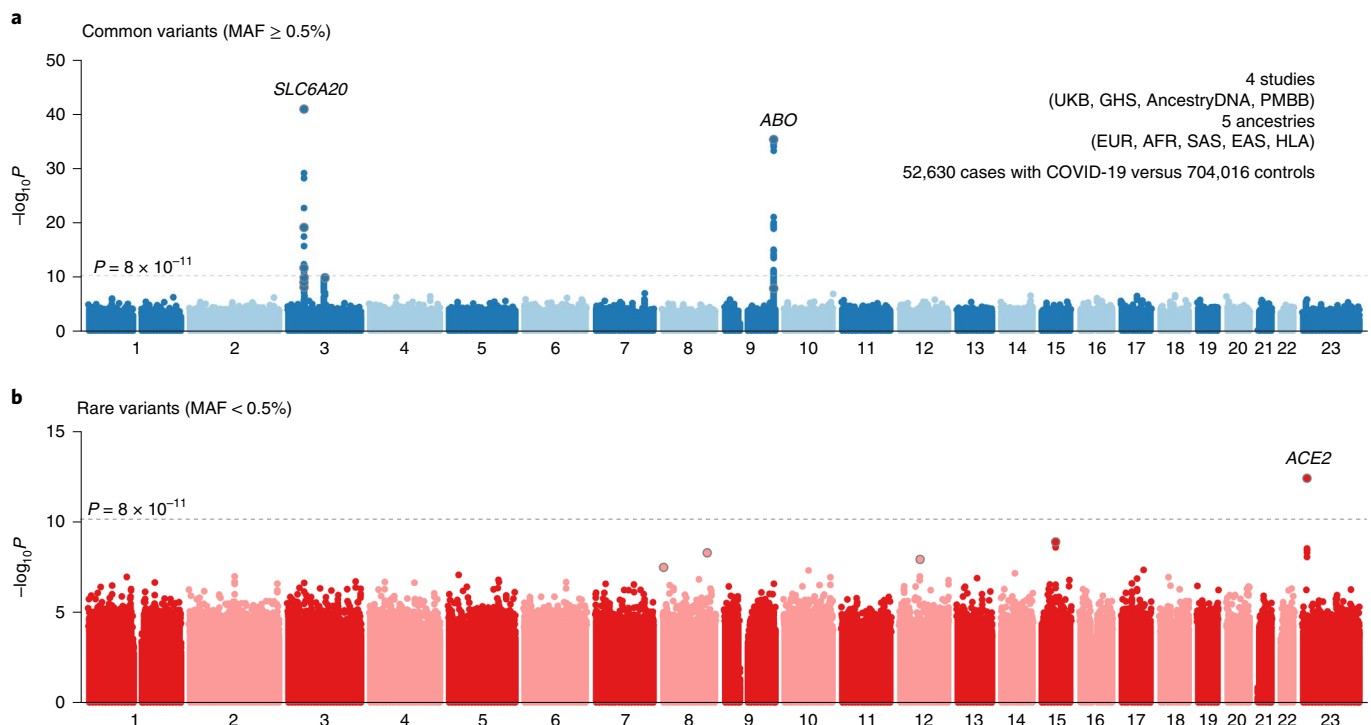

**Fig. 1 | Summary of association results from a GWAS meta-analysis of risk of infection** ($n = 52,630$ COVID-19 positive cases, $n = 704,016$ COVID-19 negative or unknown controls). **a**, Results for common variants (MAF ≥ 0.5%). **b**, Results for rare variants (MAF < 0.5%).

(95% CI = −1.18 to −0.57, linear regression test $P = 2.7 \times 10^{-8}$; Fig. 3a). When considering raw, prenormalized *ACE2* expression levels, rs190509934:C was associated with a 37% reduction in expression relative to non-carriers (Fig. 3b). There was no association with the expression of 8 other nearby genes (within 500 kilobases (kb), with detectable expression in our dataset) after accounting for multiple testing. These results are consistent with rs190509934:C lowering *ACE2* expression, which in turn confers protection from SARS-CoV-2 infection.

In addition to its role in viral infections, the normal physiological role of ACE2 involves its hydrolysis and clearance of angiotensin II, a vasoconstrictive peptide that can lead to higher vascular tone or blood pressure[19]. Therefore, we investigated if rs190509934:C was associated with higher systolic blood pressure in the UKB study but found no significant association (Beta = 0.009 s.d. units, $P = 0.56$; Supplementary Table 11). There was a trend for higher blood pressure among carriers of ultrarare coding variants in *ACE2* that are predicted to be full loss of function (Beta = 0.219 s.d. units, $P = 0.09$; Supplementary Table 11) and which were assayed through exome sequencing[20]. These results need to be confirmed in larger datasets but suggest that *ACE2* loss of function may modestly increase blood pressure. This should be considered if ACE2 blockade is to be developed for COVID-19 treatment, although pharmacological inhibition of ACE2 in such a setting would be expected to be short term and elevations in blood pressure could be managed with antihypertensives. Of note, *ACE2* expression in the airways was reported to be higher in smokers and patients with chronic obstructive pulmonary disease (COPD)[21] and to increase with age[22]. Collectively, these observations and our genetic findings are consistent with the hypothesis that ACE2 levels play a key role in determining COVID-19 risk.

**Replication of previously reported associations.** As noted, our GWAS also identified associations at several loci reported in previous

GWAS of COVID-19 outcomes. To explore previously reported signals in detail, we first attempted to replicate 8 independent associations (linkage disequilibrium (LD) $r^2 < 0.05$) with disease risk (Supplementary Table 5) reported in 3 recent GWAS[9–11] that included >1,000 cases (Supplementary Table 6). After accounting for multiple testing, 6 variants had a significant ($P < 0.0012$) and directionally consistent association in at least 1 of our 5 disease risk analyses (Supplementary Table 12): rs73064425:T in *LZTFL1* (published OR = 2.14; strongest in our analysis of cases with severe COVID-19 versus COVID-19-negative or unknown controls; MAF = 7%, OR = 1.58, $P = 2 \times 10^{-18}$); rs2531743:G near *SLC6A20* (published OR = 0.92; COVID-19-positive versus COVID-19-negative; MAF = 42%, OR = 0.94, $P = 3 \times 10^{-12}$); rs143334143:A in the major histocompatibility complex (MHC) (published OR = 1.85; COVID-19-positive versus COVID-19-negative; MAF = 7%, OR = 1.06, $P = 2 \times 10^{-4}$); rs879055593:T in *ABO* (published OR = 1.17; COVID-19-positive versus COVID-19-negative or unknown; MAF = 24%, OR = 1.10, $P = 7 \times 10^{-34}$); rs2109069:A in *DPP9* (published OR = 1.36; cases hospitalized with COVID-19 versus COVID-19-negative or unknown; MAF = 31%, OR = 1.10, $P = 3 \times 10^{-7}$); and rs2236757:A in *IFNAR2* (published OR = 1.28; cases hospitalized with COVID-19 versus COVID-19-negative or unknown; MAF = 29%, OR = 1.08, $P = 7 \times 10^{-5}$). The variants in *LZTFL1* and *SLC6A20* are located 63 kb apart at the 3p21.31 locus first reported by Ellinghaus et al.[9], which contains a core risk haplotype that includes 13 variants in high LD with each other[23]. However, in individuals of European ancestry, this haplotype block (indexed by rs35044562) is in high LD with the *LZTFL1* variant rs73064425 ($r^2 = 0.99$) but not the *SLC6A20* variant rs2531743 ($r^2 = 0.02$), indicating that these two signals—for severe COVID-19 among infected individuals and for risk of SARS-CoV-2 infection compared with individuals who did not test positive for COVID-19, respectively—are likely independent.

There was no evidence for heterogeneity in effect sizes across studies (all with $P > 0.05$; Supplementary Table 12) or ancestries

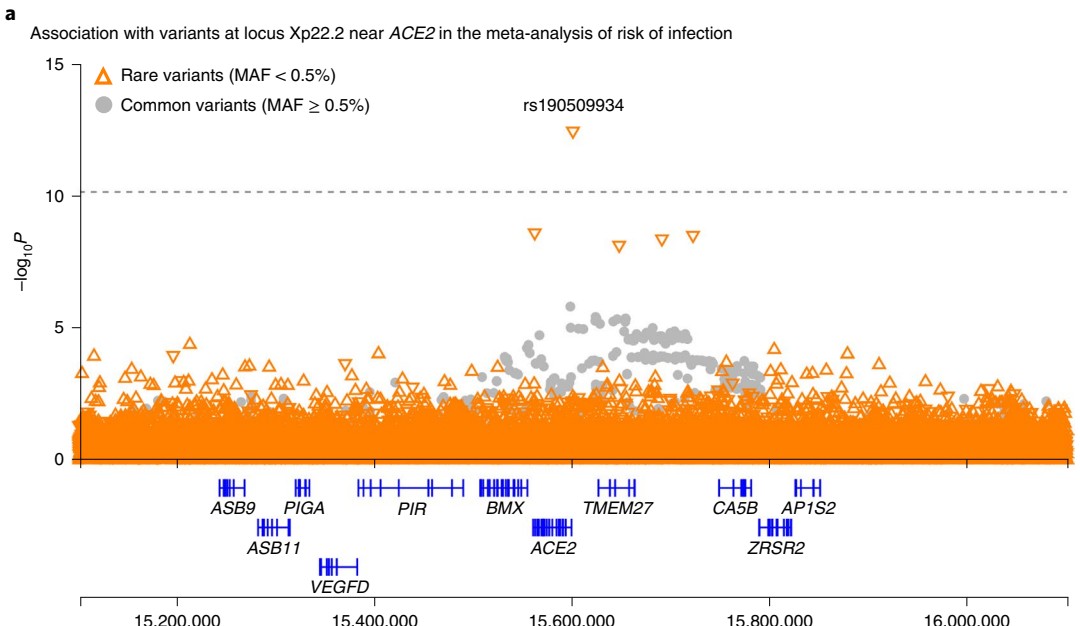

**a**
Association with variants at locus Xp22.2 near *ACE2* in the meta-analysis of risk of infection

**b**
Association between risk of infection and rs190509934:C across 12 cohorts

| Study | Number of cases RR\|RA\|AA | Number of controls RR\|RA\|AA | | OR (95% CI) | P | Alternative allele frequency |
|---|---|---|---|---|---|---|
| UKB_EUR | 14,317\|35\|2 | 420,556\|1,244\|509 | | 0.627 (0.492–0.8) | $1.7 \times 10^{-4}$ | 0.0026 |
| UKB_AFR | 616\|3\|1 | 8,721\|67\|16 | | 0.783 (0.375–1.636) | 0.516 | 0.0058 |
| UKB_SAS | 750\|12\|1 | 9,565\|217\|100 | | 0.507 (0.325–0.792) | 0.003 | 0.0219 |
| AncestryDNA_EUR | 25,306\|<100\|<100 | 113,489\|327\|<100 | | 0.569 (0.444–0.728) | $7.5 \times 10^{-6}$ | 0.002 |
| AncestryDNA_AFR | 1,622\|<100\|<100 | 5,641\|<100\|<100 | | 0.392 (0.197–0.78) | 0.008 | 0.0047 |
| AncestryDNA_SAS | <100\|<100\|<100 | 250\|<100\|<100 | | 0.359 (0.042–3.071) | 0.35 | 0.0195 |
| AncestryDNA_AMR | 3,742\|<100\|<100 | 12,281\|<100\|<100 | | 0.825 (0.443–1.537) | 0.545 | 0.0021 |
| PMBB_EUR | 40\|0\|0 | 6,993\|3\|3 | | 0.43 (0.001–187.431) | 0.786 | 0.0013 |
| PMBB_AFR | 347\|2\|0 | 8,518\|28\|18 | | 0.915 (0.286–2.928) | 0.881 | 0.0045 |
| GHS_EUR | 5,267\|9\|0 | 107,839\|261\|68 | | 0.658 (0.432–1.003) | 0.052 | 0.0019 |
| GHS_AFR | 128\|0\|0 | 3,025\|14\|11 | | 0.493 (0.126–1.921) | 0.308 | 0.0057 |
| GHS_AMR | 88\|0\|0 | 1,296\|11\|0 | | 0.304 (0.015–6.097) | 0.436 | 0.0039 |
| Meta-analysis | 52,294\|122\|8 | 698,174\|2,251\|812 | | 0.6 (0.522–0.689) | $4.5 \times 10^{-13}$ | 0.0027 |

**Fig. 2 | Association between variants near *ACE2* and risk of infection. a**, Regional association plot for locus Xp22.2 near *ACE2* in the meta-analysis of risk of infection across 14 cohorts ($n = 52,630$ COVID-19-positive cases, $n = 704,016$ COVID-19-negative or unknown controls; Supplementary Table 4). **b**, Association between risk of infection and the most significant variant at the Xp22.2 locus (rs190509934:C, MAF = 0.3%) across 12 cohorts ($n = 52,424$ COVID-19-positive cases, $n = 701,237$ COVID-19-negative or unknown controls). The variant was not tested in two cohorts due to low sample size (AncestryDNA, EAS ancestry; UKB, EAS ancestry). Associations were estimated in each cohort using Firth regression (two-sided test) as implemented in REGENIE[37], with results combined across cohorts using an inverse variance meta-analysis.

(all with $P > 0.05$; Supplementary Table 13) for any of the six variants. We also explored the possibility that the association between these six variants and COVID-19 could have been confounded by disease status for relevant comorbidities. We found that only two of the six variants were associated with a clinical risk factor: the MHC variant was associated with asthma ($P = 6.8 \times 10^{-9}$) and type 2 diabetes (T2D) ($P = 1.5 \times 10^{-5}$), while the *ABO* variant was associated with kidney

disease ($P = 1.4 \times 10^{-4}$) and T2D ($P = 9.7 \times 10^{-5}$; Supplementary Table 8). Importantly, however, for both variants the association with COVID-19 was essentially unchanged after adjusting for the associated clinical risk factors (MHC: OR = 1.09 versus OR = 1.08; *ABO*: OR = 1.08 versus OR = 1.07; Supplementary Table 14). Therefore, we conclude that the association between the six variants and COVID-19 is unlikely to be explained by these underlying comorbidities.

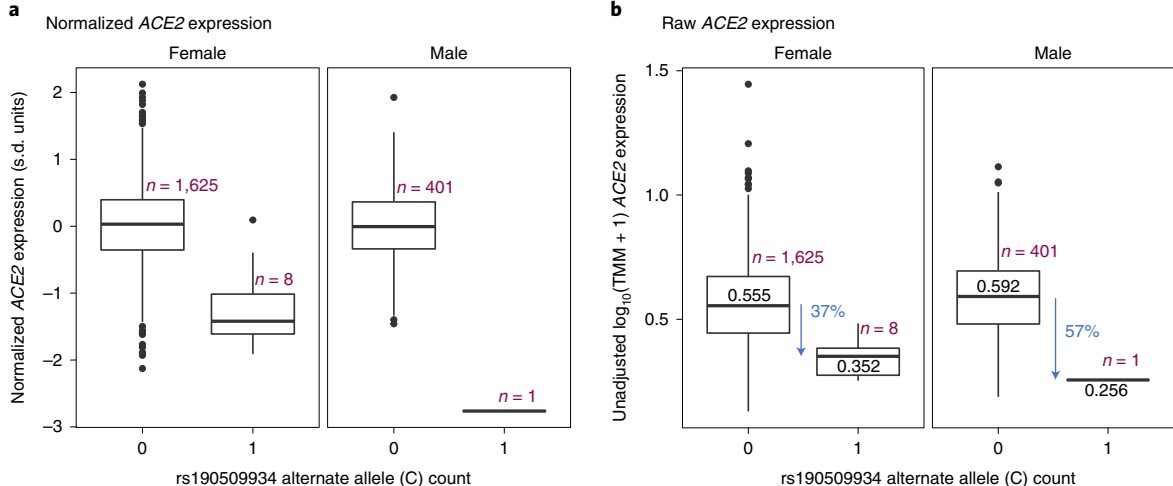

**Fig. 3 | Association between rs190509934:C and *ACE2* expression in liver measured in the GHS study (*n* = 2,035 individuals). a**, Association with normalized gene expression levels. **b**, Association with raw gene expression levels. The box plots show the median (center line), lower and upper quartiles (box boundaries), minimum and maximum (whiskers) and samples >1.5 s.d. units from the mean (individual data points).

**Associations with disease severity among cases with COVID-19.** We then investigated which replicated variants were associated with severity among cases with COVID-19. Among the 6 replicated variants (in/near *LZTFL1*, *SLC6A20*, MHC, *ABO*, *DPP9* and *IFNAR2*), 4 were significantly (*P* < 0.05) associated with worse outcomes among infected individuals (in/near *LZTFL1*, MHC, *DPP9* and *IFNAR2*), while those in *ABO* and near *SLC6A20* were not associated with COVID-19 severity (Extended Data Fig. 1 and Supplementary Table 15). Collectively, these results highlight four variants associated with both COVID-19 risk and worse disease outcomes, including respiratory failure and death. These variants may be used to identify individuals at risk of severe COVID-19 and guide the search for genes involved in the pathophysiology of COVID-19.

Next, we evaluated whether variants identified by the COVID-19 HGI, a large worldwide effort to identify genetic risk factors for COVID-19, could augment this set of four disease severity variants. The latest HGI analyses[12] include data from 49,562 individuals infected with SARS-CoV-2 and use >1.7 million individuals with no record of infection as controls (Supplementary Table 16). To identify additional variants associated with severity, we started with variants associated with the phenotype 'reported infection' (infected versus no record of infection) which, despite the sample overlap between the HGI and our analyses, was statistically independent from severity among infected individuals because infection status (positive cases versus negative or unknown controls) is uncorrelated with hospitalization status once infected (hospitalized versus non-hospitalized cases). We found that two variants were nominally associated with the risk of severe disease among cases (rs11919389 near *RPL24*, *P* = 0.029 and rs1886814 near *FOXP4*, *P* = 0.018; Supplementary Table 16), suggesting that these loci also modulate disease severity after infection with SARS-CoV-2.

**Likely effector genes of variants associated with COVID-19.** Collectively, our association analyses highlighted six common variants identified in previous GWAS or by the HGI—in/near *LZTFL1*, MHC, *DPP9*, *IFNAR2*, *RPL24* and *FOXP4*—that are associated with COVID-19 as well as disease severity among cases. To help identify genes that might underlie the observed associations, we searched for functional protein-coding variants (missense or predicted loss of function) in high LD (*r*² > 0.80) with each variant. We found eight

functional variants in five genes (Supplementary Table 17): *IFNAR2*, a cytokine receptor component in the antiviral type 1 IFN pathway, which is activated by SARS-CoV-2 and is dysregulated in cases with severe COVID-19[14,24]; *CCHCR1*, a P-body protein associated with cytoskeletal remodeling and messenger RNA turnover[25,26]; *TCF19*, a transcription factor associated with hepatitis B[27]; and *C6orf15* and *PSORS1C1*, two functionally uncharacterized genes in the MHC. These data indicate that the variants identified may have functional effects on these five genes.

We then asked if any of the 6 sentinel variants colocalized (that is, were in high LD, *r*² > 0.80) with published sentinel expression quantitative trait loci (eQTLs) across 52 studies (considering eQTLs associated with gene expression at a *P* < 2.5 × 10⁻⁹ in the original studies; Supplementary Table 10), specifically focusing on 114 genes in *cis* (±500 kb). We found colocalization with sentinel eQTLs for eight genes (Supplementary Table 18): *SLC6A20* (eQTLs from lung), a proline transporter that binds the host SARS-CoV-2 receptor, ACE2[28]; *NXPE3* (esophagus), a gene of unknown function; *SENP7* (blood), a SUMO-specific protease that promotes IFN signaling and that in mice is essential for innate defense against herpes simplex virus 1 infection[29]; *IFNAR2* and *TCF19* (multiple tissues), both discussed above; *LST1* (blood), an immunomodulatory protein that inhibits lymphocyte proliferation[30] and is upregulated in response to bacterial ligands[31]; *HLA-C* (adipose tissue), a natural killer cell ligand, which is associated with HIV infection[32] and autoimmunity[33]; and *IL10RB* (multiple tissues), a pleiotropic cytokine receptor associated with persistent hepatitis B and autoimmunity[34,35]. Collectively, analysis of missense variation and eQTL catalogs suggests 12 potential effector genes in COVID-19 loci (*ACE2*, *C6orf15*, *CCHCR1*, *HLA*-C, *IFNAR2*, *IL10RB*, *LST1*, *NXPE3*, *PSORS1C1*, *SENP7*, *SLC6A20* and *TCL19*), although functional studies are required to confirm these predictions.

**Using GRS to predict severe disease.** Next, we proceeded to evaluate if common genetic variants can help identify individuals at high risk of severe COVID-19 once infected with SARS-CoV-2. To this end, we created a weighted GRS for individuals with a record of SARS-CoV-2 infection and then compared the risk of hospitalization (hospitalized versus non-hospitalized cases) and severe disease (severe versus non-hospitalized cases) between those with a high GRS and all other cases, after adjusting for established risk factors. We considered different approaches to select variants for inclusion

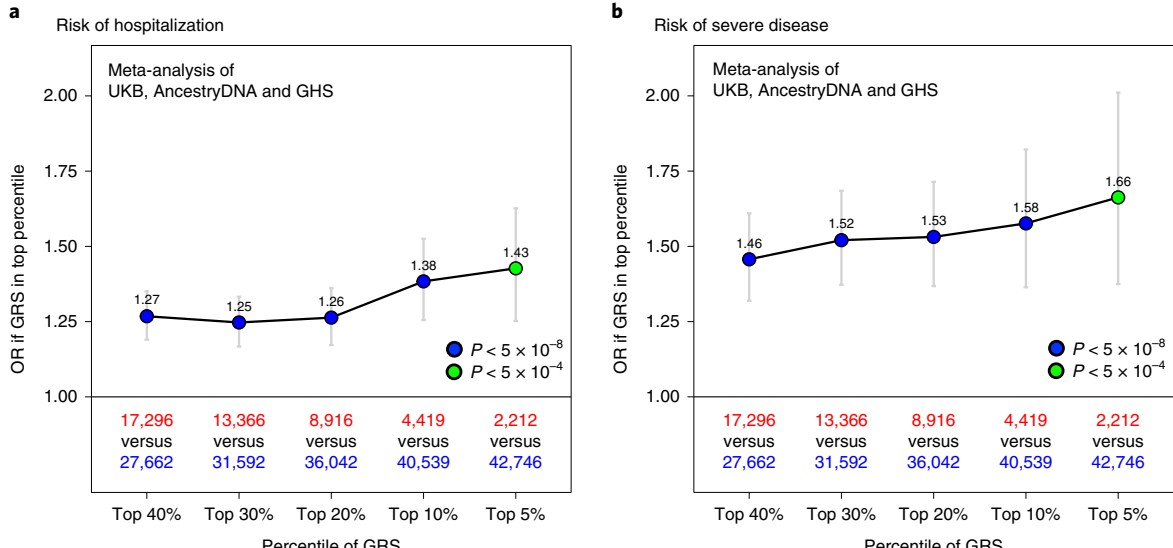

**Fig. 4 | Association between a 6-SNP GRS and risk of hospitalization and severe disease among cases with COVID-19 of European ancestry.**
**a**, Association between a high GRS and risk of hospitalization. The risk of hospitalization among cases is shown for individuals in the top GRS percentile, agnostic to the number of clinical risk factors present. The association was tested in three studies separately (AncestryDNA, UKB and GHS) using logistic regression (two-sided test), with established risk factors for COVID-19 included as covariates (Methods). Results were then meta-analyzed across studies (combined $n = 44,958$ cases with COVID-19, including 6,138 hospitalized). **b**, Association between a high GRS and risk of severe disease. The association was tested as described above in three studies separately (AncestryDNA, UKB and GHS). Results were then meta-analyzed across studies (combined $n = 44,958$ cases with COVID-19, including 1,940 with severe disease). $n$ in red: number of cases with COVID-19 in the top GRS percentile. $n$ in blue: number of cases with COVID-19 in the rest of the population. Data are presented as OR ± 95% CIs. Association statistics, including exact $P$ values, are shown in Supplementary Table 20.

in the GRS. First, we reasoned that variants most informative for prediction of severe disease were those associated with worse disease outcomes among infected individuals; thus, this was the approach taken for our primary GRS analysis. Of all published genetic risk factors for COVID-19, only one variant was associated with worse outcomes among infected individuals at $P < 5 \times 10^{-8}$ in our analysis (rs73064425 in *LZTFL1*) but this likely reflects low power due to the small number of patients with severe illness that were available for analysis. To address this limitation, we also included in the GRS five additional variants (in/near MHC, *DPP9*, *IFNAR2*, *RPL24* and *FOXP4*) that (1) had an association with risk of infection at $P < 5 \times 10^{-8}$ in published GWAS or by the HGI; and (2) were associated with worse disease outcomes among infected individuals in our data (Supplementary Tables 15 and 16), albeit at the suggestive level with current sample sizes. The combination of a genome-wide significant association with risk of infection in previous GWAS and a suggestive association with worse outcomes among infected individuals in the current analysis minimizes the chance that these loci represent false positive associations for disease severity. Of note, we did not include in the GRS five additional variants discovered by the HGI for risk of hospitalization or severe disease (Supplementary Table 16) because the HGI analysis for those two phenotypes was not statistically independent from our analysis of disease outcomes among infected individuals (due to sample overlap). To calculate the GRS, the weights used for each of the six variants corresponded to the effect size (log of the OR) reported in previous GWAS. $P$ values reported in this section were obtained from a logistic regression test (Methods), unless otherwise noted.

When considering cases with COVID-19 of European ancestry ($n = 44,958$), we found that having a high GRS (top 10%) was associated with a 1.38-fold increased risk of hospitalization (95% CI = 1.26–1.53, $P = 6 \times 10^{-11}$; Fig. 4a) and 1.58-fold increased risk of severe disease (95% CI = 1.36–1.82, $P = 7 \times 10^{-10}$; Fig. 4b). In other

ancestries, a high GRS also appeared to predict risk of hospitalization—including among individuals of African ancestry ($n = 2,598$, 1.70-fold risk for high GRS, 95% CI = 1.03–2.81, $P = 0.038$), Hispanic or Latin American ancestry ($n = 3,752$, 1.56-fold risk, 95% CI = 1.00–2.43, $P = 0.05$) and South Asian ancestry ($n = 760$, 1.42-fold risk, 95% CI = 0.72–2.82, $P = 0.32$; Supplementary Table 19). A similar pattern was observed in non-European ancestries for risk of severe disease, although sample sizes were considerably smaller (Supplementary Table 20).

We then compared the effect of the GRS between individuals with and without established risk factors for severe COVID-19. In Europeans of both the AncestryDNA and UKB studies, we found that a high GRS (top 10%) was associated with risk of severe disease both among individuals with and without established clinical risk factors for severe COVID-19 (Fig. 5). In the meta-analysis of the two studies, a high GRS was associated with a 1.65-fold (95% CI = 1.39–1.96, $P = 1 \times 10^{-8}$) and 1.75-fold (95% CI = 1.28–2.40, $P = 4 \times 10^{-4}$) higher risk of severe disease, respectively among individuals with ($n = 22,045$) and without ($n = 22,913$) established risk factors (Supplementary Table 21), with no evidence for heterogeneity of GRS effect with clinical risk factor status ($P = 0.30$). Similar results were observed for risk of hospitalization (1.35-fold versus 1.39-fold; Supplementary Table 21 and Extended Data Fig. 2). We also performed this stratified analysis in individuals of Hispanic or Latin American ancestry (but not other ancestries due to small sample size) and found that a high GRS was associated with higher risk of severe disease in individuals with ($n = 1,341$; OR = 3.35, 95% CI = 1.56–7.21, $P = 0.002$) but not without ($n = 2,411$; OR = 0.88, 95% CI = 0.19–4.07, $P = 0.873$) clinical risk factors (Extended Data Fig. 3).

Next, we performed sensitivity analyses to understand the extent to which the GRS composition affected the association results described above. First, we expanded the GRS to include all 12 variants

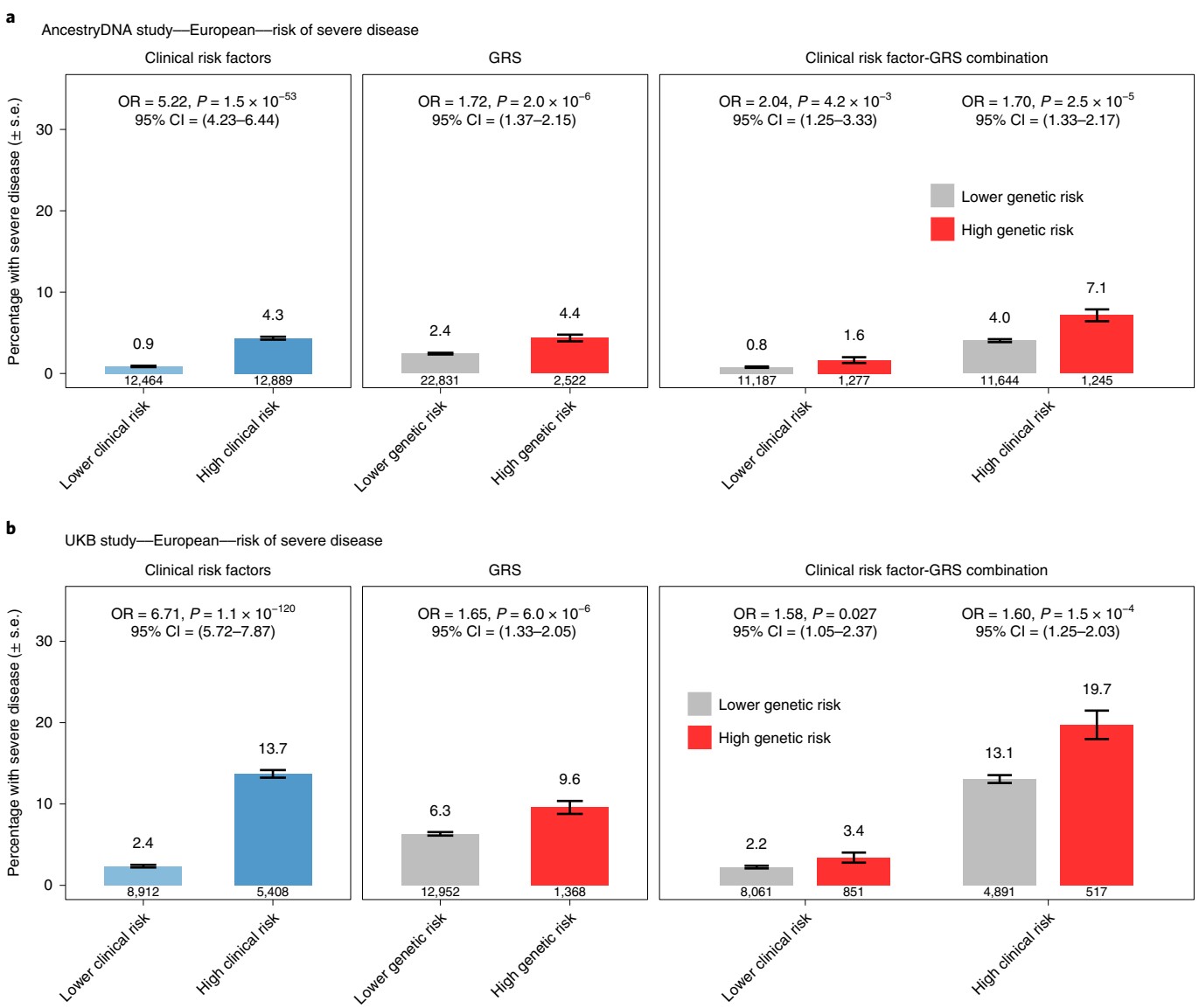

**a** AncestryDNA study—European—risk of severe disease

**b** UKB study—European—risk of severe disease

**Fig. 5 | Association between a 6-SNP GRS and risk of severe disease among cases with COVID-19 of European ancestry after stratifying by the presence of clinical risk factors. a**, Rate of severe disease in the AncestryDNA study ($n = 25{,}353$ cases with COVID-19, including 667 with severe disease). **b**, Rate of severe disease in the UKB study ($n = 14{,}320$ cases with COVID-19, including 951 with severe disease). High genetic risk (red bars): top 10% of the GRS. Low genetic risk (gray bars): bottom 90% of the GRS (that is, all other cases with COVID-19). The association between risk of severe disease and risk factors (for example, clinical risk factors) was estimated using logistic regression (two-sided test). Data are presented as the percentage of individuals with severe disease ± s.e.

reported to associate with the risk of COVID-19 in previous GWAS (8 variants) and by the HGI (4 new variants associated with reported infection). We found that associations between the 12-SNP GRS and both risk of hospitalization and severe disease were similar to those obtained with the 6-SNP GRS (Extended Data Fig. 4). For example, using the 12-SNP GRS, we found that cases with COVID-19 in the top 10% of genetic risk had a 1.38-fold (95% CI = 1.26–1.52, $P = 4 \times 10^{-11}$) and 1.64-fold (95% CI = 1.43–1.90, $P = 6 \times 10^{-12}$) higher risk of severe disease, compared to 1.38-fold and 1.58-fold, respectively obtained with the 6-SNP GRS (above). Second, we expanded the GRS to include a larger set of variants associated with risk of infection but this resulted in weaker associations when compared to the 6-SNP GRS (Extended Data Fig. 5). Overall, these results suggest that a GRS calculated using variants associated with disease risk and severity can potentially be used

to identify cases with COVID-19 at high risk of developing poor disease outcomes.

To formally address this possibility, we assessed the value of using the 6-SNP GRS to predict the risk of severe disease in addition to demographic and clinical risk factors. For this analysis, each study was split 50:50 into a training set, which was used to estimate associations between disease severity and demographic, clinical and genetic risk factors, and a validation set, where risk scores were calculated based on the effect estimates from the training set and then used to predict disease severity (Methods). We found that the ability to predict disease severity improved somewhat when the 6-SNP GRS was added to a baseline model that considered only age and sex, with the area under the receiving operator characteristic curve (AUC) improving by 0.7% in the AncestryDNA study and 0.5% in the UKB study (Fig. 6). This magnitude of

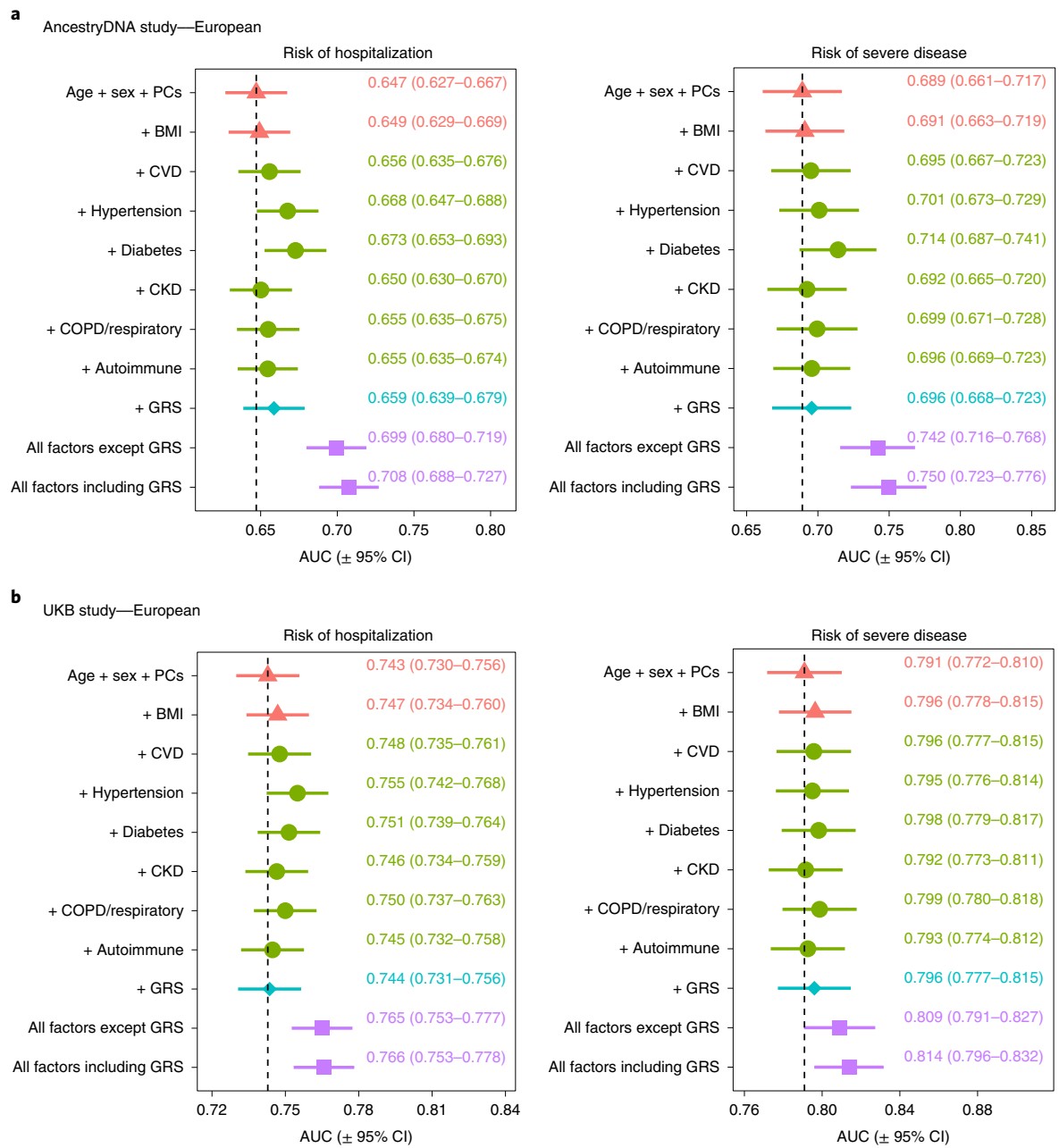

**Fig. 6 | Prediction of risk of hospitalization and severe disease among cases with COVID-19 of European ancestry based on demographic, clinical and genetic risk factors.** We tested the extent to which information on genetic risk (specifically the 6-SNP GRS) could help predict risk of hospitalization and severe disease in addition to demographic and clinical risk factors. **a**, Results for the AncestryDNA study ($n = 25,353$ cases with COVID-19). **b**, Results for the UKB study ($n = 14,320$ cases with COVID-19). Each study was split 50:50 intro training and validation sets, with prediction accuracy in the validation set summarized in each plot by the AUC. Data are presented as the AUC ± 95% CI. The vertical dashed line shows the AUC for the baseline model (age + sex + PCs).

improvement in the AUC was comparable to that observed with some clinical risk factors individually, such as cardiovascular disease (CVD) (0.6% and 0.5%, respectively in AncestryDNA and UKB) and respiratory disease (1% and 0.8%, respectively). Similar results were observed when the 6-SNP GRS was added to a model that considered all non-genetic risk factors (Fig. 6), with the AUC for disease severity improving by 0.8% and 0.5%, respectively in the AncestryDNA and UKB studies. Overall, in our analyses, age and sex were the strongest predictors of poor outcomes in individuals with COVID-19 and an elevated GRS enabled a modest

improvement in predictions similar to that contributed by individual clinical risk factors.

## Discussion
In summary, we performed a GWAS including 756,646 individuals aggregated across 4 cohorts and used both clinical and self-reported phenotypes to define risk and severity groups for COVID-19. Our analysis identified a new association between a rare variant near the *ACE2* gene that decreases expression of the SARS-CoV-2 receptor and COVID-19 risk. This finding provides human genetic support

for the hypothesis that *ACE2* expression plays a key role in SARS-CoV-2 infection and may constitute an attractive therapeutic target for prevention of COVID-19. We also confirmed six common variant associations with risk of infection and further showed that four of these variants modulate disease severity among cases. Lastly, we demonstrated that a GRS based on common variants validated in this study modestly improves the prediction of poor disease outcomes among individuals with COVID-19.

The following caveats should be considered when interpreting the results from this study. First, our study had greater power to identify associations with disease risk than with severity outcomes, given the relatively small sample size for the latter. Second, there was phenotypic heterogeneity among cases with COVID-19 and controls and associated risk factors across our studies. One likely reason for this is that survey respondents from the AncestryDNA study were enriched for healthier individuals and cases with milder COVID-19 compared to participants of the UKB, GHS and PMBB studies, who were ascertained in clinical settings and so were enriched for hospitalized cases and cases with severe COVID-19. Other sources of heterogeneity may include regional and temporal availability of COVID-19 testing and the inability to control for viral exposure among controls. While our meta-analysis collectively spans a broad phenotypic spectrum, these individual differences may account for variability in results across reported studies. Third, we used expression levels measured in the liver to assess the impact of the *ACE2* risk variant on gene expression. The liver is not the most disease-relevant tissue to assess *ACE2* expression but we note that *cis* eQTLs are often shared across tissues[18,36] and so our findings are likely predictive of decreased *ACE2* expression in other tissues. Fourth, the association between GRS and risk of severe disease was strongest in European individuals of the AncestryDNA ($OR = 1.72$, $P = 2 \times 10^{-6}$) and UKB ($OR = 1.65$, $P = 6 \times 10^{-6}$) studies when compared to the smaller GHS study ($OR = 1.03$, $P = 0.877$). The lower effect size in the latter may be due to differences in ascertainment of COVID-19-positive cases, as discussed above, or stochastic, given the smaller sample size. We also noted that the impact of the GRS on risk of hospitalization was attenuated in comparison to severe disease, which may be a reflection of the weighting schema for the variants comprising the score; the four largest GRS weights were derived from an analysis of critically ill individuals[10].

To date, SARS-CoV-2 has infected >230 million people globally, disproportionately affecting older, male individuals and those of non-European ancestry or with underlying cardiovascular and respiratory comorbidities with severe COVID-19 and death. Host genetic analysis, primarily of hospitalized cases and clinical data, have uncovered over a dozen loci associated with increased odds of severe COVID-19[12]. Our approach of coupling human genetics with both electronic health records (EHRs) and self-reported COVID-19 data has strengthened our knowledge of COVID-19 host genetics and uncovered an additional COVID-19 locus in *ACE2*. Further analysis, including additional rare variants, may further elucidate the host genetic contribution to COVID-19 and sequelae.

## Online content

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

## RGC Management and Leadership Team

**Gonçalo Abecasis[1], Aris Baras[1,6], Michael Cantor[1], Giovanni Coppola[1], Andrew Deubler[1], Aris Economides[1], Katia Karalis[1], Luca A. Lotta[1], John D. Overton[1], Jeffrey G. Reid[1], Katherine Siminovitch[1] and Alan Shuldiner[1]**

## Sequencing and Lab Operations

**Christina Beechert[1], Caitlin Forsythe[1], Erin D. Fuller[1], Zhenhua Gu[1], Michael Lattari[1], Alexander Lopez[1], John D. Overton[1], Maria Sotiropoulos Padilla[1], Manasi Pradhan[1], Kia Manoochehri[1], Thomas D. Schleicher[1], Louis Widom[1], Sarah E. Wolf[1] and Ricardo H. Ulloa[1]**

## Clinical Informatics

**Amelia Averitt[1], Nilanjana Banerjee[1], Michael Cantor[1], Dadong Li[1], Sameer Malhotra[1], Deepika Sharma[1] and Jeffrey Staples[1]**

## Genome Informatics

**Xiaodong Bai[1], Suganthi Balasubramanian[1], Suying Bao[1], Boris Boutkov[1], Siying Chen[1], Gisu Eom[1], Lukas Habegger[1], Alicia Hawes[1], Shareef Khalid[1], Olga Krasheninina[1], Rouel Lanche, Adam J. Mansfield[1], Evan K. Maxwell[1], George Mitra[1], Mona Nafde, Sean O'Keeffe[1], Max Orelus[1], Razvan Panea[1], Tommy Polanco, Ayesha Rasool[1], Jeffrey G. Reid, William Salerno[1], Jeffrey C. Staples[1], Kathie Sun[1] and Jiwen Xin[1]**

## Analytical Genomics and Data Science

**Gonçalo Abecasis[1], Joshua Backman[1], Amy Damask[1,5], Lee Dobbyn[1], Manuel Allen Revez Ferreira[1], Arkopravo Ghosh[1], Christopher Gillies[1], Lauren Gurski[1], Eric Jorgenson[1], Hyun Min Kang[1], Michael Kessler[1], Jack A. Kosmicki[1,5], Alexander Li[1], Nan Lin[1], Daren Liu[1], Adam Locke[1], Jonathan Marchini[1], Anthony Marcketta[1], Joelle Mbatchou[1], Arden Moscati[1], Charles Paulding[1], Carlo Sidore[1], Eli Stahl[1], Kyoko Watanabe[1], Bin Ye[1], Blair Zhang[1] and Andrey Ziyatdinov[1]**

## Therapeutic Area Genetics

**Ariane Ayer[1], Aysegul Guvenek[1], George Hindy[1], Giovanni Coppola[1], Jan Freudenberg[1], Jonas Bovijn, Julie E. Horowitz[1], Katherine Siminovitch[1], Kavita Praveen[1], Luca A. Lotta[1], Manav Kapoor[1], Mary Haas[1], Moeen Riaz[1], Niek Verweij[1], Olukayode Sosina[1], Parsa Akbari[1], Priyanka Nakka[1], Sahar Gelfman,**

**Sujit Gokhale[1], Tanima De[1], Veera Rajagopal[1], Alan Shuldiner[1], Bin Ye[1], Gannie Tzoneva[1] and Juan Rodriguez-Flores[1]**

**RGC Biology**

**Shek Man Chim[1], Valerio Donato[1], Aris Economides[1], Daniel Fernandez, Giusy Della Gatta[1], Alessandro Di Gioia[1], Kristen Howell[1], Katia Karalis[1], Lori Khrimian, Minhee Kim[1], Hector Martinez[1], Lawrence Miloscio[1], Sheilyn Nunez, Elias Pavlopoulos[1] and Trikaldarshi Persaud[1]**

**Research Program Management & Strategic Initiatives**

**Esteban Chen[1], Marcus B. Jones[1], Michelle G. LeBlanc[1], Jason Mighty[1], Lyndon J. Mitnaul[1], Nirupama Nishtala[1] and Nadia Rana[1]**

## Methods

**Ethical statement.** *UKB study*. Ethical approval for the UKB study was previously obtained from the North West Centre for Research Ethics Committee (no. 11/NW/0382). The work described in this study was approved by the UKB under application no. 26041.

*GHS study*. Approval for the DiscovEHR analyses was provided by the GHS institutional review board under project no. 2006-0258.

*AncestryDNA study*. All data for this research project was from individuals who provided prior informed consent to participate in AncestryDNA's Human Diversity Project, as reviewed and approved by our external institutional review board, Advarra (formerly Quorum). All data were de-identified before use.

*PMBB study*. Appropriate consent was obtained from each participant regarding the storage of biological specimens, genetic sequencing and genotyping, and access to all available EHR data. This study was approved by the institutional review board of the University of Pennsylvania and complied with the principles set out in the Declaration of Helsinki. Written informed consent was obtained for all study participants.

**Participating studies.** *AncestryDNA COVID-19 research study*. AncestryDNA customers over the age of 18, living in the USA, who had consented to the research, were invited to complete a survey assessing COVID-19 outcomes and other demographic information. These included SARS-CoV-2 swab and antibody test results, COVID-19 symptoms and severity, brief medical history, household and occupational exposure to SARS-CoV-2 and blood type. A total of 163,650 AncestryDNA survey respondents were selected for inclusion in this study. Respondents selected for this study included all individuals with a positive COVID-19 test together with age- and sex-matched controls. DNA samples were genotyped on an Illumina array containing 730,000 SNPs. Sample quality control (QC) involved removing individuals with discordant sex (based on reported and genetically determined sex) and those with <98% sample call rate, as described previously[38] Variant QC involved removing array variants with a difference in allele frequency >0.1 between any pair of array versions used, as well as variants with a call rate <98%. Genotype data for variants not included in the array were then inferred using imputation to the Haplotype Reference Consortium (HRC) reference panel. Briefly, samples were imputed to HRC v.1.1, which consists of 27,165 individuals and 36 million variants. The HRC reference panel does not include indels; consequently, indels are not present in the imputed data. We determined best-guess haplotypes with Eagle v.2.4.1 and performed imputation with Minimac4 v.1.0.1. We used 1,117,080 unique variants as input; 8,049,082 imputed variants were retained in the final dataset. Variants with a Minimac4 $r^2 < 0.30$ were filtered from the analysis.

*GHS*. The GHS MyCode Community Health Initiative is a health system-based cohort from central and eastern Pennsylvania (USA) with ongoing recruitment since 2006[39]. A subset of 144,182 MyCode participants sequenced as part of the GHS-Regeneron Genetics Center DiscovEHR partnership were included in this study. Information on COVID-19 outcomes was obtained through the GHS COVID-19 registry. Patients were identified as eligible for the registry based on relevant laboratory results and International Classification of Diseases, Tenth Revision (ICD-10) diagnosis codes. Patient charts were then reviewed to confirm the COVID-19 diagnoses. The registry contains data on outcomes, comorbidities, medications, supplemental oxygen use, and intensive care unit admissions. DNA from participants was genotyped on either the Illumina Infinium OmniExpressExome or Global Screening Array (GSA) and imputed to the TOPMed reference panel (stratified by array) using the TOPMed Imputation Server. Before imputation, we retained variants that had a MAF ≥ 0.1%, missingness <1% and Hardy–Weinberg equilibrium test $P > 10^{-15}$. After imputation, data from the Infinium OmniExpressExome and GSA datasets were merged for subsequent association analyses, which included an Infinium OmniExpressExome/GSA batch covariate, in addition to other covariates described below.

*PMBB study*. The PMBB contains approximately 70,000 study participants, all recruited through the University of Pennsylvania Health System (UPHS). Participants donate blood or tissue and allow access to EHR information[40]. The PMBB participants with COVID-19 infection were identified through the UPHS COVID-19 registry, which consists of quantitative PCR (qPCR) results of all patients tested for SARS-CoV-2 infection within the health system. We then used EHRs to classify patients with COVID-19 into hospitalized and severe (ventilation or death) categories. DNA genotyping was performed with the Illumina GSA and imputation performed using the TOPMed reference panel as described for the GHS study.

*UKB study*. We studied the host genetics of SARS-CoV-2 infection in participants of the UKB study, which took place between 2006 and 2010 and includes approximately 500,000 adults aged 40–69 at recruitment. In collaboration with the UK health authorities, the UKB has made available regular updates on COVID-19 status for all participants, including results from four main data types: qPCR test for SARS-CoV-2; anonymized EHRs; primary care; and death registry data. We report results based on phenotype data downloaded on the 4 January 2021 and excluded from the analysis 28,547 individuals with a death registry event before 2020. DNA samples were genotyped as described previously[41] using the Applied Biosystems UK BiLEVE Axiom Array ($n = 49,950$) or the closely related (95% variant overlap) Applied Biosystems UKB Axiom Array ($n = 438,427$). Genotype data for variants not included in the arrays were inferred using the TOPMed reference panel, as described above.

**COVID-19 phenotypes used for the genetic association analyses.** We grouped participants from each study into three broad COVID-19 disease categories (Supplementary Table 1): (1) positive, that is, those with a positive qPCR or serology test for SARS-CoV-2 or with a COVID-19-related ICD-10 code (U07), hospitalization or death; (2) negative, that is, those with only negative qPCR or serology test results for SARS-CoV-2 and with no COVID-19-related ICD-10 code (U07), hospitalization or death; and (3) unknown, that is, those with no qPCR or serology test results and no COVID-19-related ICD-10 code (U07), hospitalization or death. We then used these broad COVID-19 disease categories, in addition to hospitalization and disease severity information, to create seven COVID-19-related phenotypes for genetic association analyses, as detailed in Supplementary Table 3.

SARS-CoV-2 infection status (positive, negative or unknown) was determined based on a qPCR test for SARS-CoV-2 in the UKB, GHS and PMBB studies and self-reported results for qPCR or serology test for SARS-CoV-2 in the AncestryDNA study.

Hospitalization status (positive, negative or unknown) was determined based on the COVID-19-related ICD-10 codes U071, U072 and U073 in variable 'diag_icd10' (table 'hesin_diag') in the UKB study, self-reported hospitalization due to COVID-19 in the AncestryDNA study and medical records in the GHS and PMBB studies.

Disease severity status (severe (ventilation or death) or not severe) was determined in the UKB study based on: (1) respiratory support ICD-10 code Z998 in variable 'diag_icd10' (table 'hesin_diag'); (2) the following respiratory support ICD-10 codes in variable 'oper4' (table 'hesin_oper'): E85, E851, E852, E853, E854,E855, E856, E858, E859, E87, E871, E872, E873, E874, E878, E879, E89, X56, X561, X562, X563, X568, X569, X58, X581, X588 and X589; or (3) the COVID-19-related ICD-10 codes U071, U072 and U073 in cause of death (variable 'cause_icd10' in table 'death_cause'). In the AncestryDNA study, disease severity was determined based on self-reported ventilation or need for supplementary oxygen due to COVID-19. In the GHS and PMBB studies, it was determined based on ventilator or high-flow oxygen use.

For association analysis in the AncestryDNA study, we excluded from the COVID-19 unknown group individuals who had (1) a first-degree relative who was COVID-19-positive or (2) flu-like symptoms.

**Genetic association analyses.** Association analyses in each study were performed using the genome-wide Firth logistic regression test implemented in REGENIE V2.0.1 (ref. [37]). In this implementation, Firth's approach is applied when the $P$ value from the standard logistic regression score test is below 0.05. We included in step 1 of REGENIE (that is, prediction of individual trait values based on the genetic data) directly genotyped variants with an MAF > 1%, <10% missingness, Hardy–Weinberg equilibrium test $P > 1 \times 10^{-15}$ and LD pruning (1,000 variant windows, 100 variant sliding windows and $r^2 < 0.9$). The association model used in step 2 of REGENIE included as covariates age, age[2], sex, age-by-sex and the first 10 ancestry-informative principal components (PCs) derived from the analysis of a stricter set of LD-pruned (50 variant windows, 5 variant sliding windows and $r^2 < 0.5$) common variants from the array (imputed for the GHS study) data.

Within each study, association analyses were performed separately for five different continental ancestries defined based on the array data: African (AFR), Hispanic or Latin American (HLA; originally referred to as 'AMR' by the 1000 Genomes Project; a subsequent study recommended the use of HLA to refer to this ancestral group[42]); European (EUR); and South Asian (SAS). We determined continental ancestries by projecting each sample onto reference PCs calculated from the HapMap3 reference panel. Briefly, we merged our samples with HapMap3 samples and kept only SNPs in common between the two datasets. We further excluded SNPs with MAF < 10%, genotype missingness >5% or Hardy–Weinberg equilibrium test $P < 10^{-5}$. We calculated PCs for the HapMap3 samples and projected each of our samples onto those PCs. To assign a continental ancestry group to each non-HapMap3 sample, we trained a kernel density estimator (KDE) using the HapMap3 PCs and used the KDEs to calculate the likelihood of a given sample belonging to each of the five continental ancestry groups. When the likelihood for a given ancestry group was >0.3, the sample was assigned to that ancestry group. When two ancestry groups had a likelihood >0.3, we arbitrarily assigned AFR over EUR, HLA over EUR, HLA over EAS, SAS over EUR and HLA over AFR. Samples were excluded from analysis if no ancestry likelihoods were >0.3 or if more than three ancestry likelihoods were >0.3.

Results were subsequently meta-analyzed across studies and ancestries using an inverse variance-weighted fixed-effects meta-analysis.

**Identification of putative targets of GWAS variants based on colocalization with eQTLs.** We identified as a likely target of a sentinel GWAS variant any gene for which a sentinel eQTL colocalized (that is, had an LD $r^2 > 0.80$) with the sentinel GWAS variant. That is, we only considered genes for which there was strong LD between a sentinel GWAS variant and a sentinel eQTL, which reduces the chance of spurious colocalization. Sentinel eQTLs were defined across 174 published datasets (Supplementary Table 10), considering only eQTLs associated with gene expression in *cis* ($\pm 1$ Mb) at a conservative $P < 2.5 \times 10^{-9}$ threshold as described previously[43]. We did not use statistical approaches developed to distinguish colocalization from shared genetic effects because these have very limited resolution at high LD levels ($r^2 > 0.80$) (ref. [44]).

**Gene expression analysis in participants of the GHS study.** For a subset of individuals from the GHS study ($n = 2,035$, ascertained through the Geisinger Bariatric Surgery Clinic), RNA was extracted from liver biopsies conducted during bariatric surgery to evaluate liver disease. Individuals had class 3 obesity (BMI > 40 kg m$^{-2}$) or class 2 obesity (BMI 35–39 kg m$^{-2}$) with an obesity-related comorbidity (for example, T2D, hypertension, sleep apnea, non-alcoholic fatty liver disease). RNA libraries were prepared using poly(A) extraction and then sequenced with 75-bp paired-end reads with two 10-bp index reads on the Illumina NovaSeq 6000 on S4 flow cells. RNA-seq data were then analyzed using the GTEx v.8 workflow[18], using STAR v.2.7.3a (ref. [45]) and rnaSeqQC v.1.2 (Code availability), except that GENCODE v.32 was used in lieu of v.26. Briefly: (1) raw expression counts were normalized with trimmed mean of M values (TMM) as implemented in edgeR v.3.13 (ref. [46]); (2) a rank-based inverse normal transformation was applied to the normalized expression values; (3) PC analysis was performed on data from 25,078 genes with transcripts per million > 0.1 in >20% samples to identify latent factors accounting for variation in gene expression; (4) gene expression levels were adjusted for the top 100 PCs to improve power to identify *cis*-regulatory effects. The association between adjusted *ACE2* expression and the imputed genotypes of rs190509934 was then tested using linear regression, with the following variables included as covariates: age, age$^2$, four ancestry-informative PCs, steatosis status, fibrosis status, diabetes status and BMI at the time of bariatric surgery.

**GRS analysis of COVID-19 hospitalization and severity.** First, in each study (AncestryDNA, GHS, UKB and PMBB), we created a GRS for each COVID-19-positive individual based on variants that were reported to associate with risk of COVID-19 in previous GWAS and that we (1) independently replicated (except variants identified by the HGI) and (2) found to be associated with COVID-19 severity outcomes. We used as weights the effect (Beta) reported in previous GWAS (Supplementary Table 5). Second, we ranked individuals with COVID-19 based on the GRS and created a new binary GRS predictor by assigning each individual to a high (top 5%) or low (rest of the population) percentile group. Third, for studies with >100 hospitalized cases, we used logistic regression to test the association between the binary GRS predictor and risk of hospitalization (hospitalized cases versus all other cases), including as covariates age, sex, age-by-sex interaction and ten ancestry-informative PCs. In addition to age and sex, we included as additional covariates established clinical risk factors for COVID-19 that are outlined in the Emergency Use Authorisation treatment guidelines for casirivimab and imdevimab: BMI; chronic kidney disease (CKD); diabetes; immunosuppressive disease; COPD or other chronic respiratory disease; CVD; and hypertension. We repeated the association analysis (1) using different percentile cutoffs for the GRS (5, 10, 20, 30 and 40%) and (2) to test the association with disease severity (severe cases versus all other cases). We then stratified COVID-19 cases by clinical risk (high versus lower) and evaluated the association between the top 10% by GRS (that is, high genetic risk) and risk of hospitalization or severe disease. The stratified analyses were performed with logistic regression, with sex and ancestry-informative PCs included as covariates. High clinical risk was defined as any one of the following: (1) age ≥65; (2) BMI ≥ 35; (3) CKD, diabetes or immunosuppressive disease; (4) age ≥55 and presence of COPD/other chronic respiratory disease, CVD or hypertension.

In populations with >100 hospitalized cases, we also evaluated the impact of the GRS relative to other non-genetic risk factors associated with increased risk of hospitalization and severe disease (for example, COPD, diabetes). The datasets were randomly split 50:50 into training and test datasets. In the training dataset, a logistic regression model with age, sex and ancestry covariates was fitted. The coefficients for age and sex from this model were then used to calculate a risk score in the other half of the population, which was fitted in a second model along with ancestry covariates. From this model, the AUC from a receiver operating characteristic curve (and 95% CI) was estimated. The process was repeated iteratively, adding other demographic and clinical risk factors one at a time to the baseline model with age, sex and ancestry covariates. Models were then fitted with just the baseline model plus GRS, all factors except GRS and a final model with all demographic/clinical risk factors plus the GRS.

**Statistics and reproducibility.** No statistical method was used to predetermine sample size. Individuals were excluded for the following reasons: if they were not assigned to one of the five continental ancestry groups based on principal component analysis (Methods), had previously passed away before January 2020 (near the beginning of the COVID-19 pandemic), had an unknown COVID-19

status but did have confirmed cases in their household or if the continental ancestry group had fewer than 25 cases and 25 controls (Methods). The experiments were not randomized. The investigators were not blinded to allocation during the experiments and outcome assessment. Unless otherwise noted, the association *P* values reported in this manuscript are from (1) Firth (disease traits) or linear (quantitative traits) regression tests performed in REGENIE for GWAS and (2) logistic regression, for the GRS analyses.

**Reporting Summary.** Further information on research design is available in the Nature Research Reporting Summary linked to this article.

## Data availability

All genotype–phenotype association results reported in this study are available for browsing using the Regeneron Genetic Center (RGC) COVID-19 Results Browser (https://rgc-covid19.regeneron.com). Data access and use is limited to research purposes in accordance with the terms of use (https://rgc-covid19.regeneron.com/terms-of-use). Gene expression levels derived from the liver RNA-seq data for *ACE2* and the eight nearby genes analyzed in this study, as well as genotypes for the *ACE2* variant associated with the risk of SARS-CoV-2 infection (rs190509934), are provided in Supplementary Data 1.

## Code availability

REGENIE v.2.0.1 can be accessed at https://github.com/rgcgithub/regenie. The GWAS analyses were performed with REGENIE using automated pipelines. An R script that exemplifies how the genetic risk score analyses were performed is available at https://doi.org/10.5281/zenodo.5700998 and https://doi.org/10.5281/zenodo.5748168. R can be found at https://www.r-project.org/. rnaSeqQC is available from GitHub (https://github.com/oicr-gsi/rnaSeqQC).

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

## Acknowledgements

This research was conducted using the UKB Resource (project no. 26041). The PMBB is funded by a gift from the Smilow family, the National Center for Advancing Translational Sciences of the National Institutes of Health under CTSA award no. UL1TR001878 and the Perelman School of Medicine at the University of Pennsylvania. We thank the participants and investigators of the FinnGen study. We thank the AncestryDNA customers who voluntarily contributed information in the COVID-19 survey.

## Author contributions

J.A.K., A.M. and M.A.R.F. designed and conducted the GWAS. J.A.K. performed the meta-analysis. J.E.H., J.A.K., A.M., G.R.A. and M.A.R.F. interpreted the results. A.D. performed the GRS analysis and interpretation. J.E.H., J.A.K., D. Sharma, N.B., A.Y., S.B., F.S.P.K., K.S., M.N.C. and M.A.R.F. assembled and created the phenotypes. R.L., E.M. and L.H. developed the COVID-19 browser. A.M., D. Sun, J.D.B., J. Mbatchou, K.W., L.G. and J. Marchini developed the pipelines. J. Mbatchou and J. Marchini developed REGENIE. J.A.K., G.H.L.R., A.E.J., M.V.C., J.B.L., D.S.P., S.C.K., X.B., H.G., A. Baltzell, A.R.G., C.O.D., S.R.M., R.P., A.J.M., D.A.T., M.Z., A.H.L., S.E.M., L.D., E.S., A.V., G.S., M.D.R., M.J., S.B., W.J.S., A.R.S., D.J.R., T.M., J.D.O., D.J.C., E.L.H., J.G.R., C.A.B., A. Baras, G.R.A. and M.A.R.F. provided the data and reagents. J.E.H., J.A.K., A.D., A.E.L., J. Marchini, A. Baras, G.R.A. and M.A.R.F. supervised the research. J.E.H., J.A.K., A.D., A.M., G.R.A. and M.A.R.F. wrote the manuscript. All authors contributed to and

approved the final manuscript. All authors: contributed to securing the funding, study design and oversight; reviewed the final version of the manuscript; performed and were responsible for sample genotyping and exome sequencing; conceived and were responsible for laboratory automation, sample tracking and the library information management system; were responsible for the development and validation of the clinical phenotypes used to identify study participants and (when applicable) controls; performed and were responsible for the analysis needed to produce exome and genotype data; provided the computing infrastructure development and operational support; provided variant and gene annotations and their functional interpretation of variants and conceived and were responsible for creating, developing and deploying the analysis platforms and computational methods used to analyze the genomic data; developed the statistical analysis plans; contributed to the quality control of the genotype and phenotype files and the generation of the analysis-ready datasets; developed the statistical genetics pipelines and tools and use thereof in the generation of the association results; contributed to the quality control of the review and the interpretation of the results and generated and formatted the results to create the manuscript figures; contributed to the development of the study design and analysis plans and the quality control of the phenotype definitions; quality-controlled, reviewed and interpreted the association results; developed the in vivo and in vitro experimental biology and interpretation, contributed to the management and coordination of all research activities, planning and execution and managed the review of the project.

## Competing interests

J.E.H., J.A.K., A.D., D. Sharma, N.B, A.Y., A.M., R.L., E.M., X.B., D. Sun, F.S.P.K., J.D.B., C.O.D., A.J.M., D.A.T., A.H.L., J. Mbatchou, K.W., L.G., S.E.M, H.M.K., L.D., E.S., M.J., S.B., K.S, W.J.S., A.R.S., A.E.L., J. Marchini, J.D.O., L.H., M.N.C., J.G.R., A. Baras, G.R.A. and M.A.R.F. are current and/or former employees and/or stockholders of RGC or Regeneron Pharmaceuticals. G.H.L.R., M.V.C., D.S.P., S.C.K. A. Baltzell, A.R.G., S.R.M., R.P., M.Z., K.A.R., E.L.H. and C.A.B. are current and/or former employees of AncestryDNA and may hold equity in AncestryDNA. The other authors declare no competing interests.

## Additional information

**Extended data** is available for this paper at https://doi.org/10.1038/s41588-021-01006-7.

**Correspondence and requests for materials** should be addressed to Manuel A. R. Ferreira.

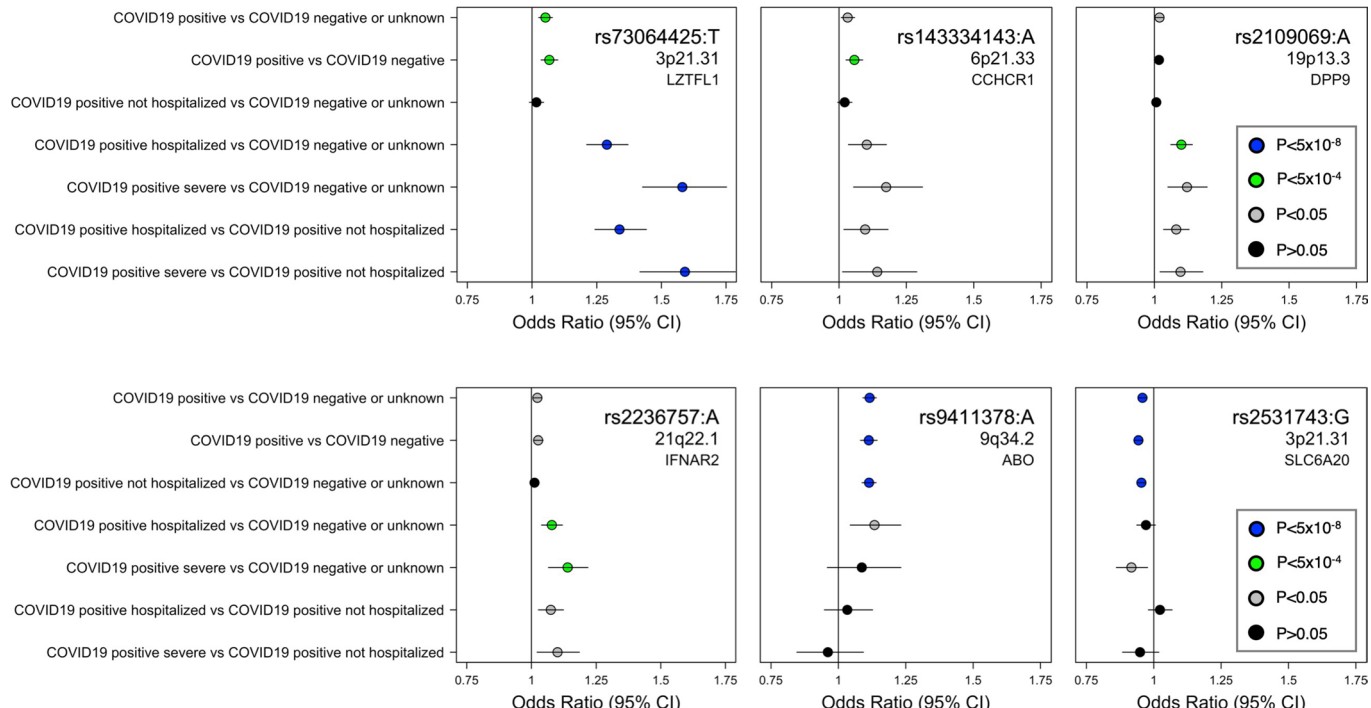

**Extended Data Fig. 1 | Comparison of effect sizes across COVID-19 risk and severity outcomes for six previously reported risk variants that validated in this study.** Six variants were reported to associate with risk of COVID-19 in previous studies and replicated in our analysis. Of these, four variants also associated with disease severity among COVID-19 cases (in/near *LZTFL1*, *CCHCR1*, *DPP9* and *IFNAR2*), whereas two variants did not (in *ABO* and *SLC6A20*). Sample size for each of the seven phenotypes is shown in Supplementary Table 3. Data are presented as odds ratio +/− 95% confidence interval.

**a   AncestryDNA study – European – Risk of hospitalization**

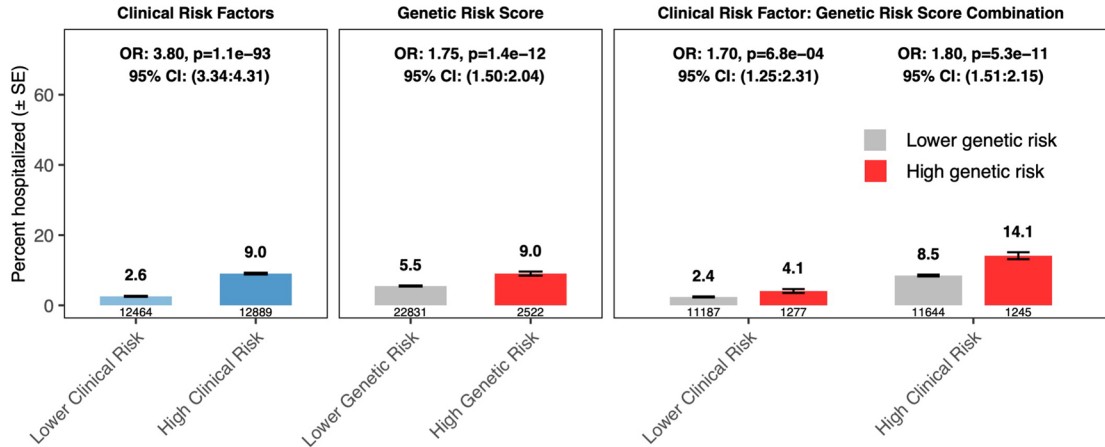

**b   UK Biobank study – European – Risk of hospitalization**

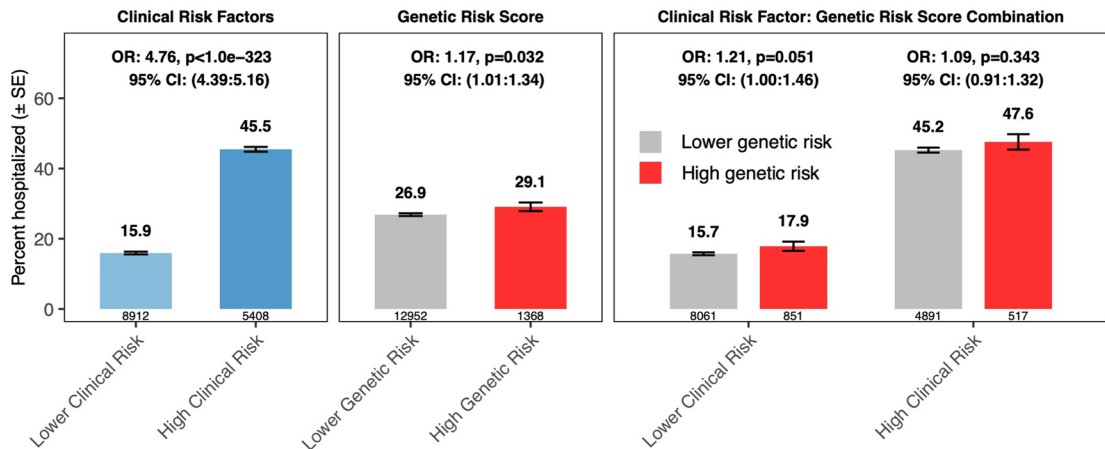

**Extended Data Fig. 2 | Association between a 6-SNP genetic risk score (GRS) and risk of hospitalization among COVID-19 cases of European ancestries after stratifying by the presence of clinical risk factors. a**, Rate of hospitalization in the AncestryDNA study (n=25,353 COVID-19 cases, including 1,484 hospitalized). **b**, Rate of hospitalization in the UK Biobank study (n=14,320 COVID-19 cases, including 3,878 hospitalized). High genetic risk (red bars): top 10% of the GRS. Low genetic risk (grey bars): bottom 90% of the GRS (that is all other COVID-19 cases). Data are presented as percent of individuals hospitalized +/- standard error (SE).

**a**  AncestryDNA study – Hispanic or Latin American – Risk of hospitalization

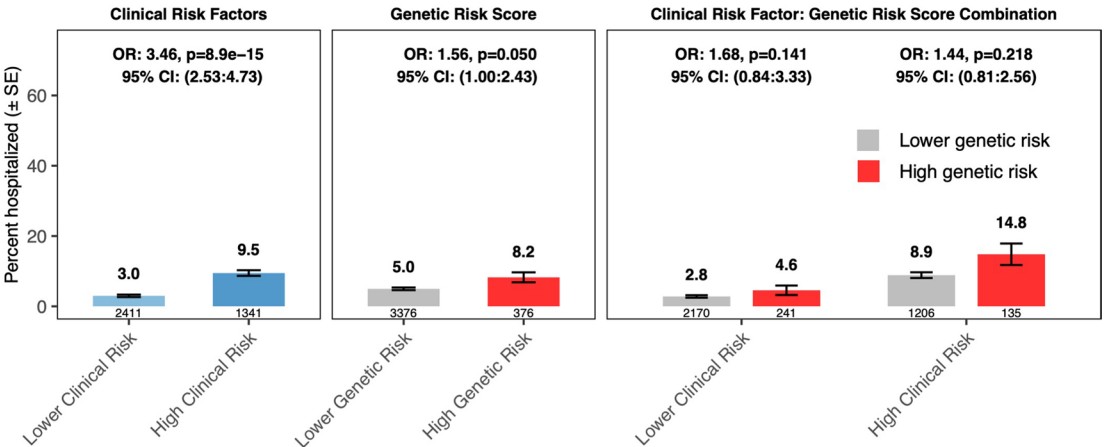

**b**  AncestryDNA study – Hispanic or Latin American – Risk of severe disease

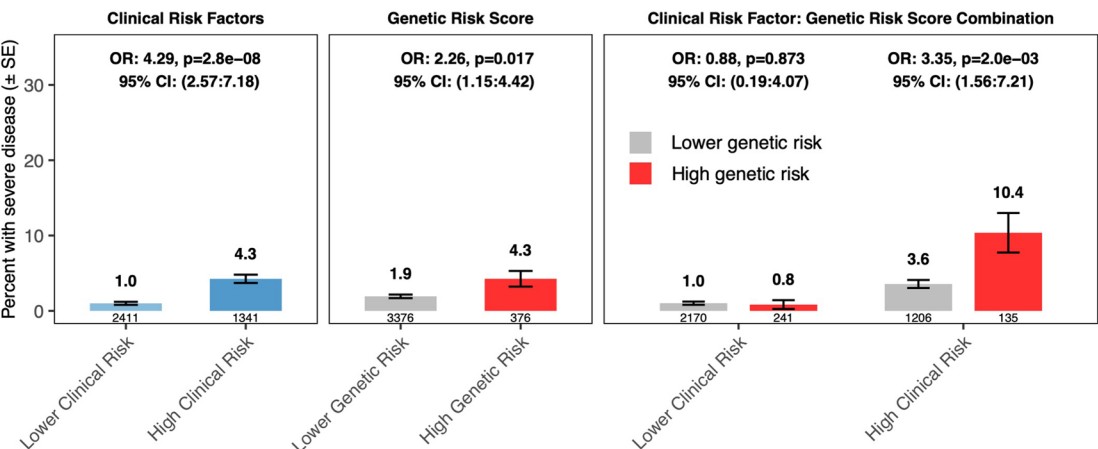

**Extended Data Fig. 3 | Association between a 6-SNP genetic risk score (GRS) and risk of hospitalization and severe disease among COVID-19 cases of Hispanic or Latin American ancestries (n = 3,752). a**, Rate of hospitalization. **b**, Rate of severe disease. High genetic risk (red bars): top 10% of the GRS. Low genetic risk (grey bars): bottom 90% of the GRS (that is all other COVID-19 cases). Data are presented as percent of individuals hospitalized (**a**) or with severe disease (**b**) ± standard error (SE).

**a** **Risk of hospitalization**

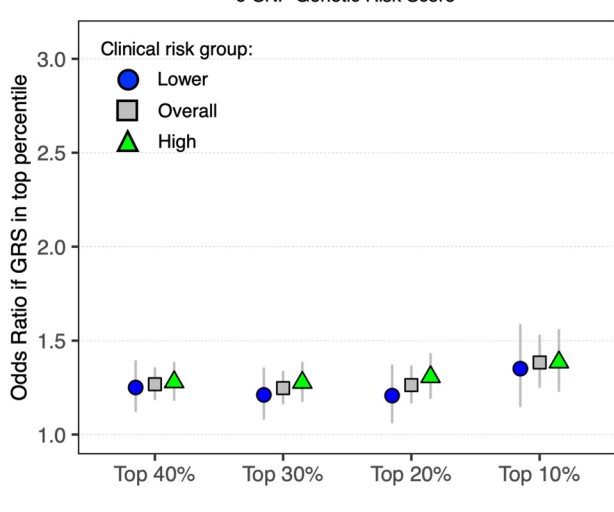

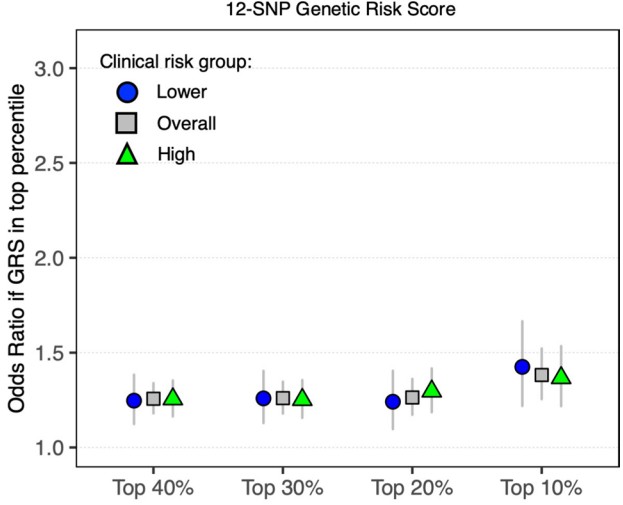

**b** **Risk of severe disease**

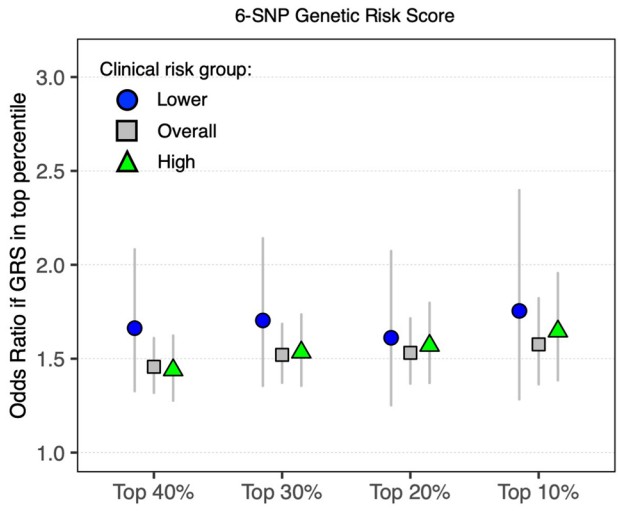

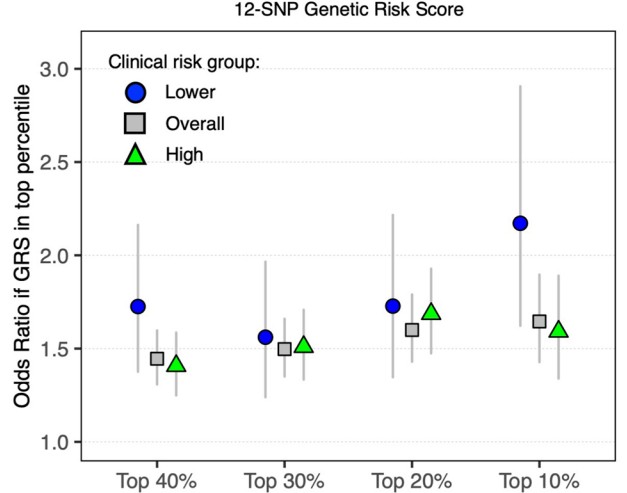

**Extended Data Fig. 4 | Association between a 6- and 12-SNP genetic risk score (GRS) and risk of hospitalization and severe disease among COVID-19 cases of European ancestries. a**, Associations with risk of hospitalization (n = 44,958 COVID-19 cases). **b**, Associations with risk of severe disease (n = 39,673). To evaluate if the association between the GRS and worse disease outcomes was dependent on the list of variants selected for analysis, we compared results between GRS calculated using different sets of variants. We considered a GRS calculated using: the six variants that were reported in previous GWAS of COVID-19 and that we further showed were associated with risk of hospitalization or severe disease among COVID-19 cases (four variants in/near *LZTFL1*, MHC, *DPP9* and *IFNAR2*, see Extended Data Fig. 1; and two variants discovered by the HGI in/near *RPL24* and *FOXP4*, see Supplementary Table 16). Analyses were performed separately in the UK Biobank, AncestryDNA and GHS studies (risk of hospitalization only) after stratifying COVID-19 cases by the presence of clinical risk factors, considering individuals with lower clinical risk (blue circles), high clinical risk (green triangles) or all individuals (grey squares). Association results were then meta-analyzed across studies. Data are presented as odds ratio +/− 95% confidence interval.

**a**  **AncestryDNA study**

**b**  **UK Biobank study**

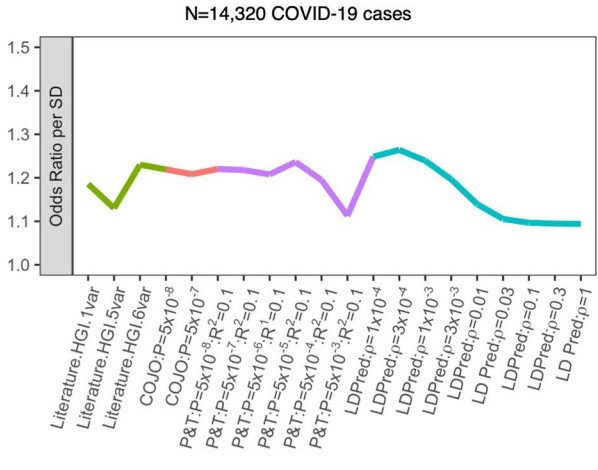

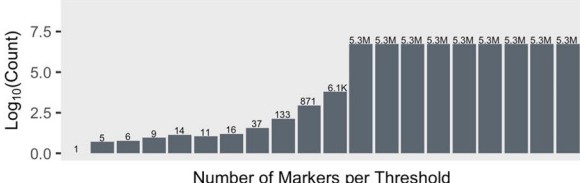
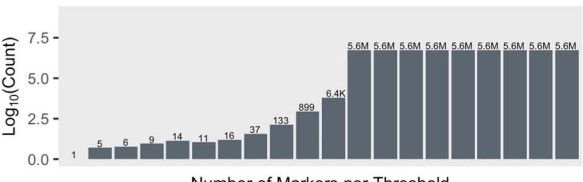

**Extended Data Fig. 5 | Association between risk of severe disease among COVID-19 cases of European ancestries and genetic risk scores (GRS) determined based on different criteria. a**, Association results in the AncestryDNA study (n = 25,353 COVID-19 cases). **b**, Association results in the UK Biobank study (n = 14,320 COVID-19 cases). In each study, we compared GRS based on (i) variants that were reported in the literature and validated in this study (Literature.HGI.1var: rs73064425 in *LZTFL1*; Literature.HGI.5var: variants from our 6-SNP model, with the exception of rs73064425 in *LZTFL1*; Literature.HGI.6var: all six variants from our 6-SNP model; in green); and variants associated with the risk of infection phenotype reported by the HGI and obtained through (ii) approximate conditional analysis using GCTA-COJO, considering two association *P*-value thresholds (5 x 10$^{-8}$ and 5 x 10$^{-7}$; in orange); (iii) pruning and thresholding (P&T), using different association P-value and LD $r^2$ thresholds (in purple); and (iv) the LDpred approach[47], considering different $\rho$ parameters (in teal).

# Reporting Summary

Nature Research wishes to improve the reproducibility of the work that we publish. This form provides structure for consistency and transparency in reporting. For further information on Nature Research policies, see our Editorial Policies and the Editorial Policy Checklist.

## Statistics

For all statistical analyses, confirm that the following items are present in the figure legend, table legend, main text, or Methods section.

| n/a | Confirmed | |
|---|---|---|
| ☐ | ☒ | The exact sample size (*n*) for each experimental group/condition, given as a discrete number and unit of measurement |
| ☐ | ☒ | A statement on whether measurements were taken from distinct samples or whether the same sample was measured repeatedly |
| ☐ | ☒ | The statistical test(s) used AND whether they are one- or two-sided *Only common tests should be described solely by name; describe more complex techniques in the Methods section.* |
| ☐ | ☒ | A description of all covariates tested |
| ☐ | ☒ | A description of any assumptions or corrections, such as tests of normality and adjustment for multiple comparisons |
| ☐ | ☒ | A full description of the statistical parameters including central tendency (e.g. means) or other basic estimates (e.g. regression coefficient) AND variation (e.g. standard deviation) or associated estimates of uncertainty (e.g. confidence intervals) |
| ☐ | ☒ | For null hypothesis testing, the test statistic (e.g. *F*, *t*, *r*) with confidence intervals, effect sizes, degrees of freedom and *P* value noted *Give P values as exact values whenever suitable.* |
| ☒ | ☐ | For Bayesian analysis, information on the choice of priors and Markov chain Monte Carlo settings |
| ☒ | ☐ | For hierarchical and complex designs, identification of the appropriate level for tests and full reporting of outcomes |
| ☒ | ☐ | Estimates of effect sizes (e.g. Cohen's *d*, Pearson's *r*), indicating how they were calculated |

*Our web collection on statistics for biologists contains articles on many of the points above.*

## Software and code

Policy information about availability of computer code

Data collection
> Data transfer from sequencing machine to DNAnexus
> -Upload Agent v1.5.30 https://wiki.dnanexus.com/Downloads#Upload-Agent
>
> Single-sample processing, all in DNAnexus
> -Conversion of sequencing data in BCL format to FASTQ format and the assignments of paired-end sequence reads to samples based on 10-base barcodes; bcl2fastq v2.19.0 https://support.illumina.com/sequencing/sequencing_software/bcl2fastq-conversion-software.html
> -Read alignment; bwa 0.7.17 http://bio-bwa.sourceforge.net
> -Duplicate marking, stats gathering; picard v1.141 https://broadinstitute.github.io/picard/
> -SAM/BAM/CRAM file generation and manipulation; samtools v1.7 http://www.htslib.org
> -Variant calling; WeCall v1.1.2 https://github.com/Genomicsplc/wecall
> -Sequence Quality Control; FastQC 0.11.8 http://www.bioinformatics.babraham.ac.uk/projects/fastqc/
> -VCF file manipulation and index generation; bcftools v1.7 http://www.htslib.org, bgzip/tabix v1.7 http://www.htslib.org
> -Multi-threaded file compression and decompression; pigz v2.3.4 https://zlib.net/pigz/
> -haplotyping (Ancestry.com); Eagle v2.4.1 https://github.com/poruloh/Eagle
> -imputation (Ancestry.com): Minimac4 v1.01 https://github.com/statgen/Minimac4
>
> Generation of "freeze" data
> -Joint genotyping to generate project-level VCF (pVCF) files; GLnexus v0.4.0 https://github.com/dnanexus-rnd/GLnexus
> -Generation of variant representations in PLINK format; PLINK v1.90b6.21 https://www.cog-genomics.org/plink2/
> -Ancestry predictions, IBD (Identity-by-descent) estimate, and pedigree reconstruction; PLINK v1.90b6.21 https://www.coggenomics.org/plink2/, PRIMUS https://primus.gs.washington.edu/primusweb/

Data analysis
> - association testing: REGENIE v2.0.1 https://github.com/rgcgithub/regenie.

| Data analysis | - meta-analysis: METAL (2020-05-05) https://github.com/statgen/METAL.<br>- various: python v3.8 https://www.python.org/downloads/; R v4.0.4 https://cran.r-project.org |

For manuscripts utilizing custom algorithms or software that are central to the research but not yet described in published literature, software must be made available to editors and reviewers. We strongly encourage code deposition in a community repository (e.g. GitHub). See the Nature Research guidelines for submitting code & software for further information.

## Data

Policy information about availability of data

All manuscripts must include a data availability statement. This statement should provide the following information, where applicable:
- Accession codes, unique identifiers, or web links for publicly available datasets
- A list of figures that have associated raw data
- A description of any restrictions on data availability

Full genotype-phenotype association results reported in this study are freely available for browsing and download using the Regeneron Genetics Center (RGC)'s COVID-19 Results Browser (https://rgc-covid19.regeneron.com). Data access and use is limited to research purposes in accordance with the Terms of Use (https://rgc-covid19.regeneron.com/terms-of-use); and agree that any public use of the Data must cite the associated publication, disclose that the RGC was the source of the data, and make the following statement: "The Regeneron Genetics Center bears no responsibility for the analyses or interpretations of the data presented here. Any opinions, insights, or conclusions presented herein are those of the authors and not of The Regeneron Genetics Center." as well as include the RGC name and logo on any public disclosure of the Data..  Users of the data may not attempt to identify any individuals who are subjects of the Data; combine the Data with other data in a manner that could lead to identification of an individual; copy or use the Data in any manner except as expressly permitted by this Agreement; directly commercialize the Data, including but not limited to, any sale, lease, license or transfer of the Data for monetary or other commercial gain; reverse engineer, disassemble, or decompile the Data; alter or remove any proprietary notices in the Data; or use or make available the Data for any purpose that is unlawful.  For further questions on data access and restrictions, please contact rgc-covidrb@regeneron.com and we will respond in a timely manner.

# Field-specific reporting

Please select the one below that is the best fit for your research. If you are not sure, read the appropriate sections before making your selection.

☒ Life sciences          ☐ Behavioural & social sciences          ☐ Ecological, evolutionary & environmental sciences

For a reference copy of the document with all sections, see nature.com/documents/nr-reporting-summary-flat.pdf

# Life sciences study design

All studies must disclose on these points even when the disclosure is negative.

| Sample size | Sample sizes were all those available in UK Biobank, Penn Medicine Biobank, Ancestry.com, or Geisinger Health Systems as described in the text. No power calculations were performed or required in advance. |
| Data exclusions | Prior to any analysis, we established the following data exclusions: We excluded individuals that were not predicted to belong to 5 continental ancestry groups (AFR, AMR, EAS, EUR, SAS) and furthermore did not analyze sets of individuals with fewer than 25 cases and 25 controls. |
| Replication | No replication was performed.  Because we contributed our data to the Host Genetics Initiative of COVID-19, we could not use their summary statistics for replication (as our samples overlapped).  Furthermore, the main novel finding, a protective variant in ACE2, was below the minimum allele frequency (5% or 1%) used by other studies and thus was not present in external summary statistics preventing replication. |
| Randomization | We performed a GWAS, which was an observational study, and as such no process of randomization was performed or applicable here because there was no allocation of samples into experimental groups. |
| Blinding | We performed a GWAS, which was an observational study, using coded de-identified data. As such, no process of blinding to group allocation was performed or applicable here. |

# Reporting for specific materials, systems and methods

We require information from authors about some types of materials, experimental systems and methods used in many studies. Here, indicate whether each material, system or method listed is relevant to your study. If you are not sure if a list item applies to your research, read the appropriate section before selecting a response.

## Materials & experimental systems

| n/a | Involved in the study |
|---|---|
| ☒ | ☐ Antibodies |
| ☒ | ☐ Eukaryotic cell lines |
| ☒ | ☐ Palaeontology and archaeology |
| ☒ | ☐ Animals and other organisms |
| ☐ | ☒ Human research participants |
| ☒ | ☐ Clinical data |
| ☒ | ☐ Dual use research of concern |

## Methods

| n/a | Involved in the study |
|---|---|
| ☒ | ☐ ChIP-seq |
| ☒ | ☐ Flow cytometry |
| ☒ | ☐ MRI-based neuroimaging |

## Human research participants

Policy information about studies involving human research participants

| | |
|---|---|
| Population characteristics | Population characteristics can be found in Supplementary Tables 1-3. |
| Recruitment | UK Biobank recruited approximately 500,000 individuals 40-69 years of age in 2006 to 2010 by mailers to people in the UK medical system. Informed consent was obtained for all participants. AncestryDNA customers over age 18, living in the United States, and who had consented to research, were invited to complete a survey assessing COVID-19 outcomes and other demographic information including SARS-CoV-2 swab and antibody test results, COVID-19 symptoms and severity, brief medical history, household and occupational exposure to SARS-CoV-2, and blood type. Geisinger Health System (GHS). The GHS MyCode Community Health Initiative is a health system-based cohort from central and eastern Pennsylvania (USA) with ongoing recruitment since 2006. Penn Medicine BioBank (PMBB) study. PMBB study participants are recruited through the University of Pennsylvania Health System, which enrolls participants during hospital or clinic visits. Any known biases from EHR-based datasets, survey datasets, or population-based datasets may be present and applicable here to this study and it's results. |
| Ethics oversight | Ethical approval for the UK Biobank was previously obtained from the North West Centre for Research Ethics Committee (11/NW/0382). The work described herein was approved by UK Biobank under application number 26041. GHS study: approval for DiscovEHR analyses was provided by the Geisinger Health System Institutional Review Board under project number 2006-0258. AncestryDNA study: all data for this research project was from subjects who provided prior informed consent to participate in AncestryDNA's Human Diversity Project, as reviewed and approved by our external institutional review board (Pro00034516), Advarra (formerly Quorum). All data was de-identified prior to use. PMBB study: appropriate consent was obtained from each participant regarding storage of biological specimens, genetic sequencing and genotyping, and access to all available EHR data. This study was approved by the Institutional Review Board of the University of Pennsylvania and complied with the principles set out in the Declaration of Helsinki. |

Note that full information on the approval of the study protocol must also be provided in the manuscript.

