## [Peer Review File · Nature Genetics]

Peer Review Information

Manuscript Title: Genome-wide analysis provides genetic evidence that ACE2 expression influences COVID-19 risk and yields risk scores associated with severe disease

Corresponding author name(s): Dr Manuel Ferreira

Reviewer Comments & Decisions:

Decision Letter, initial version:
--

10th Mar 2021

Dear Manuel,

Your Brief Communication, "Common genetic variants identify targets for COVID-19 and individuals at high risk of severe disease" has now been seen by 2 referees. You will see from their comments copied below that while they find your work of considerable potential interest, they have raised important concerns that must be addressed. In light of these comments, we cannot accept the manuscript for publication, but would be very interested in considering a revised version that addresses these serious concerns.

In brief, both reviewers find your study to be of interest and appear open to considering a revision. We note that both highlighted the GRS as being of interest and we believe that this would be an important aspect to expand on in a revision.

However, they both also comment on technical aspects of the analysis. For example, Reviewer #1 questions why a more 'traditional' study design using discovery and validation cohorts was not performed. Both referees also point out issues with the variability in the studied populations and how these will affect the association analyses; e.g. Reviewer #1 asking about the phenotyping and Reviewer #2 making a comparable comment regarding the prevalence of comorbidities. Beyond these major comments, they also make other suggestions for improvements to the technical aspects of the analysis that we found to be clear, constructive guidance that, if acted upon, would improve the study.

We hope you will find the referees' comments useful as you decide how to proceed. If you wish to submit a substantially revised manuscript, please bear in mind that we will be reluctant to approach the referees again in the absence of revisions that fully address their comments.

To guide the scope of the revisions, the editors discuss the referee reports in detail within the team, including with the chief editor, with a view to identifying key priorities that should be addressed in revision and sometimes overruling referee requests that are deemed beyond the scope of the current study. We hope that you will find the prioritised set of referee points to be useful when revising your study.

Please do not hesitate to get in touch if you would like to discuss these issues further; I am happy to arrange a call if this would be of service.

If you choose to revise your manuscript taking into account all reviewer and editor comments, please highlight all changes in the manuscript text file. At this stage we will need you to upload a copy of the manuscript in MS Word .docx or similar editable format.

*2) If you have not done so already please begin to revise your manuscript so that it conforms to our Brief Communication format instructions, available [here](http://www.nature.com/ng/authors/article_types/index.html). Refer also to any guidelines provided in this letter.

[REDACTED]

If you wish to submit a suitably revised manuscript we would hope to receive it when ready; we appreciate that COVID-19 is a very fast-developing topic and, as such, are willing to give you and your

co-authors the time you feel is needed to improve the manuscript as outlined in the reports.

We will be happy to consider your revision so long as nothing similar has been accepted for publication at Nature Genetics or published elsewhere. Should your manuscript be substantially delayed without notifying us in advance and your article is eventually published, the received date would be that of the revised, not the original, version.

Thank you for the opportunity to review your work.

Sincerely,

Michael Fletcher, PhD
Associate Editor, Nature Genetics

ORCID: 0000-0003-1589-7087

Referee expertise: GWAS, infectious diseases.

Reviewers' Comments:

Reviewer #1:

Remarks to the Author:

Horowitz et al. presents a series of analyses related to four Covid-19 GWAS datasets, i.e. UK Biobank, Ancestry, Penn Biobank, Geisinger Health System subset (participating in the GHS-Regeneron Genetics Center DiscovEHR partnership). Key output of interest is an elaboration on GRS for severe Covid-19, two novel claims of associations at KAT7 (5q34) and TMPRSS2 (16q24.3) and a discussion related to MHC associations in Covid-19 (represented by CCHCR1 just telomeric to the class I/class III junction). The paper starts with a discussion on previously published risk loci as seen from the perspective of a trans-ethnic meta-analysis of the current four datasets.

I have the following considerations for the authors:

1) The re-use of datasets across different studies is becoming quite extensive for Covid-19 (e.g. UKBB and Pairo-Castineira in Nature; and Roberts et al. at MedRxiv). The reason for the choice of the initial

meta-analysis versus confirmatory reporting of previously published risk loci, versus a more traditional two-stage approach of discovery analysis in the previously unpublished cohorts (UPENN/GHS) with replication in other datasets (UKBB, Ancestry, others) remains blurred (beyond statistical power). The exact overlap of (and potentially study unique) cases and controls versus other studies utilizing the UKBB and Ancestry databases would be useful for future scrutiny of results and comparisons across reports. Major differences in study populations are evident between the constructed case-control sets (e.g. case control 20k vs. 654k for the C19 pos vs. C19 neg/unknown compared with case control ratio of 1k vs. 17k for C19 pos severe vs. C 19 pos not hospitalized). How do the authors consider these differences in statistical power between the various subsets of analysis may bias the overall reporting?

2) Phenotype details are only superficially reported for the two novel cohorts (the separate sources of data for UKBB and Ancestry makes the referencing for these two cohorts acceptable). This must be considerably expanded (as supplementary material). Furthermore, importantly, there is no detailed elaboration on how the clinical phenotype assignments align across the four cohorts, there are differences – e.g. how to define respiratory failure (by questionnaire? with hospitalization as a proxy? by means of respiratory support? etc.). These “phenotype substructures” (some of which remains cryptic) brings a bias in the meta-analysis which is not properly adjusted for (in comparison to population substructures, which are dealt with in standard ways for these types of studies). This problem becomes evident in the several signals reported for the chromosome 3 haplotype (LD is very strong here), sometimes with a lead signal in SLC6A20 sometimes with a signal in LZTFL1 – which does not represent independent signals as clarified by fine-mapping, rather surfaces in various case-control phenotype comparisons. The claim of number of independent loci in the first part of the manuscript thus becomes false. This must be fixed, and unless finemapping is performed, no independency of the chromosome 3 associations must be claimed.

3) On what basis do the authors claim that the MHC / HLA is an established risk locus for Covid-19 (page 4 line 71)? As far as this reviewer is aware, no consistent HLA association signal has been detected in any of the biggest published studies nor in the Covid-19 Host Genetics Initiative paper. Please clarify. Furthermore, great caution must be taken when reporting HLA associations – in general, and in trans-ethnic analysis in particular, given the huge risk of cryptic population substructures. I remain unconvinced on the validity of the CCHCR1 association reported for this reason (unless confirmatory data is provided), as well as the lack of regular clarification you would see for an HLA association (e.g. alignment with classical loci alleles, HLA-C and HLA-B in particular – including the dimorphisms relevant for KIR interactions to which the authors allude). A simple regional association plot would have been useful in clarifying this region in more detail.

4) This reviewer has major problems with the term “target” for genetic findings. Only an extreme minority of GWAS outcomes have proven their efficacy as drug targets, and this way of reporting genetic associations holds a corporate bias in the context of a study within the broader Regeneron scope. I would refer to loci, genes and variants for detected associations, title included. The referencing to antibody cocktails in the reporting of the GRS holds the same corporate bias and should be deleted (ditto for vaccine), as this is an indirect speculation and lacks direct evidence. The topic of severe Covid-19 cannot afford unfounded clinical suggestions.

5) The two novel associations arise not from the meta-analysis mentioned under above point 1, but by an expanded meta-analysis using data also from FinnGen, Ellinghaus et al. and the GenOMICC studies. Hits are noted at KAT7 and TMPRSS2 at just about genome-wide significance. Whilst certainly

of some interest, TMRSS2 in particular, I would report the loci with caution in lack of independent validation (are the authors able to bring forward such replication?), and moderate claims related to treatment of patients (e.g. suggesting that inhibition of TMRSS2 might be protective against Covid-19). Could an alternative approach to reporting the analyses seen in this paper have been the two novel cohorts separately, and then the expanded meta-analyses resulting in these two novel findings?

6) The GRS section and analysis is to this reviewer the most interesting part of the manuscript. How does GRS risk stratification perform in comparison to risk stratification by clinical risk factors?

Reviewer #2:

Remarks to the Author:

The authors present a GWAS meta-analysis of four different studies, which were further stratified by genetic ancestry. This paper represents a large amount of work, both replicating previous associations, identifying two novel putative associations, fine-mapping the locations, and assessing the information given by a genetic risk score on top of clinical risk factors. My first concern with the analyses lies in the lack of investigation around comorbidities which represent a significant source of confounding in these GWAS. The authors show significant differences in comorbidities between cases and controls, and additionally between severity levels. However, their association models do not take these differences into account. Therefore, it is unclear if the results are specific to COVID-19, or rather a GWAS of underlying comorbidities. This should be straightforward to address. My second major concern is with the use of 'susceptibility' when there is no assessment of exposure to the virus among controls. I detail these concerns and several additional major and minor points below.

MAJOR POINTS

- The authors show in Supplementary Table 2 that their cases and controls had very different levels of comorbidities, such as hypertension, CVD, T2D, CKD, asthma, and COPD. Additionally, the not hospitalized case category had lower levels of comorbidities than those who were hospitalized. These proportions also vary greatly between cohorts. For example, 44% of those hospitalized in GHS had cardiovascular disease (CVD), while only 15.9% of controls had CVD. UK Biobank had only 14.2% of hospitalized cases with CVD and 5.9% of controls. These imbalances of comorbidities, which are known independent risk factors for severe COVID-19, are not included in any of the association models for the GWAS or replication analyses. Therefore, there is potential for substantial confounding in which results measure nothing specific to COVID, but rather measure genetics related to these comorbidities. Sensitivity analyses should be conducted, as well as controlling for comorbidity burden within the regression models.
- Another major concern is the use of 'susceptibility' as an outcome. Infectious disease requires exposure to the pathogen for an individual to develop the outcome of interest (infection). As the vast majority of controls were likely not exposed during the recruitment time, the comparison for SARS-CoV-2 positive tests to negative tests or no test does not measure true susceptibility. Instead it measures both the behavior of exposure and susceptibility to infection. I suggest changing the terminology to 'risk of positive test' or something similar to distinguish this.
- There were no tests of heterogeneity presented across studies or ancestries as part of the meta-analysis. Given how heterogeneous the effect sizes look in the forest plots, it would be good to present these analyses and interpret why some effect sizes may be wildly different even despite similar sample size.
- The authors do not currently discuss possible reasons for heterogeneity in case/control status and

associated risk factors between studies. It would be beneficial for the manuscript to touch on this, as there are a large number of COVID-19 GWAS being published and no way to distinguish them and their conflicting results. These results need to be contextualized by describing the underlying study population and any benefits/limitations therein.

- The GRS and clinical risk factors analyses is important and shows the added benefit of genetics to traditional risk scores. However, there are major differences between the AncestryDNA and UK Biobank results, in which the GRS effects are significantly attenuated in UK Biobank. Additionally, the GRS is further attenuated in risk of hospitalization. The authors should discuss potential reasons for this lack of transferability, such as overfitting for AncestryDNA or differences in healthcare systems and ability to be hospitalized.
- There is no replication for the novel risk variants identified in this study. This should be done either through internal validation, or doing a lookup of the previously published GWAS that were not included in this analysis.
- In general, the manuscript currently reads like a list of results, instead of a description of the results and adequate expansion to explain why the results matter to the reader, as well as provide interpretations of the major points and potential limitations. I understand this is due to the limits of the format as a brief communication, but it would be helpful to rebalance some of the text to help the reader.

MINOR POINTS

- The authors briefly touch on the five phenotypes related to disease risk and two phenotypes. I would include some sort of flow chart or diagram to understand these within the main text if there is space. It would also be useful to put in some text explaining how these phenotypes differ in a meaningful way.
- I am assuming that "Admixed American" refers to Hispanic/Latino individuals. However, this term is confusing and is not consistent with prior literature or the standard in the field.
- Please put the actual numbers throughout the manuscript instead of <100, especially as some cohorts have this information and some just have <100.
- It would be helpful to include the effect sizes of the previously published loci and how they compare to your effect sizes in the main text. This would also help to contextualize your results which show differences in magnitude across studies.

Author Rebuttal to Initial comments

Reviewer #1:

1. The reason for the choice of the initial meta-analysis versus confirmatory reporting of previously published risk loci, versus a more traditional two-stage approach of discovery analysis in the previously unpublished cohorts (UPENN/GHS) with replication in other datasets (UKBB, Ancestry, others) remains blurred (beyond statistical power).

As the reviewer alludes to, maximizing statistical power was our main concern when we planned the two-pronged approach used to identify variants associated with severe COVID-19. First, we sought to independently replicate the association with published risk variants and then assess

their effect on severity among cases. The pros and cons of this first approach are obvious: we maximize power by focusing on a small number of reported risk variants (significant threshold can be relaxed) but, of course, this approach can only confirm published (not discover novel) associations. The second approach – meta-analysis of COVID-19 susceptibility across all datasets available to us – addressed the latter limitation. We opted for this one-stage approach because it almost always provides greater power to identify true associations than a two-stage approach (discovery followed by replication) {PMID: 16415888}.

Since our original submission, the COVID-19 Host Genetics Initiative (which we contributed to) published a preprint that describes GWAS of three COVID-19 susceptibility phenotypes with a sample size that was considerably greater than that included in our original analysis. As such, the GWAS we performed as part of approach #2 no longer provided the most powerful analysis of COVID-19 outcomes reported to date. For this reason, we revised approach #2 to include (i) only datasets for which we had access to individual-level data (UKB, GHS, UPENN and AncestryDNA); and (ii) an analysis that to our knowledge has not been reported to date: association between COVID-19 outcomes and rare variants assayed through imputation. In doing so, we identified a rare variant (MAF=0.3%) in the promoter region of ACE2 (the primary cell entry receptor for SARS-CoV-2) that strongly protects against SARS-CoV-2 infection (OR=0.60, 95% CI 0.52-0.69, P=4.5x10⁻¹³). We also show that this variant strongly reduces ACE2 expression (P=10⁻⁹) based on liver RNA-seq data for 2,067 individuals from the GHS study.

2. The exact overlap of (and potentially study unique) cases and controls versus other studies utilizing the UKBB and Ancestry databases would be useful for future scrutiny of results and comparisons across reports.

This is a great suggestion. In our manuscript, we refer to genetic associations with COVID-19 identified in four published studies:

Study	PMID	N cases	N controls	Sample overlap with our study
Ellinghaus	32558485	1,610	2,205	No known overlap
Pairo-Castineira	33307546	1,676	8,380	~1% of samples from our study were included as controls in this study
Shelton	33888907	12,972	101,268	No known overlap
HGI (freeze 5)	NA	49,562	1.7 million	~100% of samples from UKB, GHS and UPENN, and ~60% from AncestryDNA contributed to HGI, but using an earlier phenotype freeze.

The first three were used to identify eight variants associated with COVID-19 susceptibility for replication in our study. We found that six of these eight variants replicated at a Bonferroni-corrected P-value in our meta-analysis of the UKB, GHS, UPENN and AncestryDNA, totaling 52,630 cases and 704,016 controls.

From the HGI study – which included ~90% of the samples used in our analysis – we selected four novel risk variants that were identified in the analysis of the “C2” phenotype (Reported infection), as this is statistically independent from our severity phenotypes among infected individuals. We found that two of these four variants were associated with worse outcomes among infected individuals.

We have now included this information in the new Supplementary Table 6.

3. Major differences in study populations are evident between the constructed case-control sets (e.g. case control 20k vs. 654k for the C19 pos vs. C19 neg/unknown compared with case control ratio of 1k vs. 17k for C19 pos severe vs. C 19 pos not hospitalized). How do the authors consider these differences in statistical power between the various subsets of analysis may bias the overall reporting?

We agree that statistical power is expected to be greater for the five susceptibility phenotypes (total N ranging from 690K to 750K) when compared to the two severity phenotypes that were measured only in COVID-19 cases (total N~50K). As such, our study is indeed biased towards identifying more associations with disease risk than with severity outcomes. We have added this limitation to a new caveats section at the end of the manuscript.

4 and 5. Phenotype details are only superficially reported for the two novel cohorts (the separate sources of data for UKBB and Ancestry makes the referencing for these two cohorts acceptable). This must be considerably expanded (as supplementary material). Furthermore, importantly, there is no detailed elaboration on how the clinical phenotype assignments align across the four cohorts, there are differences – e.g. how to define respiratory failure (by questionnaire? with hospitalization as a proxy? by means of respiratory support? etc.). These “phenotype substructures” (some of which remains cryptic) brings a bias in the meta-analysis which is not properly adjusted for (in comparison to population substructures, which are dealt with in standard ways for these types of studies).

The seven phenotypes we analyzed were defined in all studies based on:

- I. Infection status: derived from qPCR or serology testing for SARS-CoV-2

- II. Hospitalization status: derived from ICD10 codes, medical records or self-report.
- III. Severity status: derived from ICD10 codes, medical records and cause of death.

To address the reviewer's concern, we have now added a paragraph to the Methods section that describes how infection, hospitalization and severity status was defined in each of the four studies.

6. This problem becomes evident in the several signals reported for the chromosome 3 haplotype (LD is very strong here), sometimes with a lead signal in SLC6A20 sometimes with a signal in LZTFL1 – which does not represent independent signals as clarified by fine-mapping, rather surfaces in various case-control phenotype comparisons. The claim of number of independent loci in the first part of the manuscript thus becomes false. This must be fixed, and unless finemapping is performed, no independency of the chromosome 3 associations must be claimed.

We no longer focus on the chromosome 3 signal, since this has now been extensively described in a number of publications.

7. On what basis do the authors claim that the MHC / HLA is an established risk locus for Covid-19 (page 4 line 71)? As far as this reviewer is aware, no consistent HLA association signal has been detected in any of the biggest published studies nor in the Covid-19 Host Genetics Initiative paper. Please clarify.

We focused our replication analysis on eight variants reported in previous GWAS. This list included the MHC variant originally reported by Pairo-Castineira et al (PMID 33307546) in their analysis of 1,676 critically ill COVID-19 cases from the UK (but not UK Biobank) and 8,380 controls taken from the UK Biobank study (rs143334143, OR=1.85, P=8.82E-18).

When we inspected the original association, we noticed that variants in high LD with rs143334143 (based on 1000 Genomes data) had much less significant associations in the Pairo-Castineira study when compared to rs143334143, which was unexpected. Therefore, we were concerned (as were others) that it could be a false-positive association arising from cases and controls having been genotyped separately. Nonetheless, there was no reason for excluding this variant from our replication analysis, which we reasoned would show no association if indeed this was a false-positive association. However, we found that:

- I. Variant rs143334143 was associated at $P < 0.05$ with four of our five susceptibility phenotypes (one at the appropriate Bonferroni-corrected P-value; Supp Table 12). The

exception was phenotype COVID-19 positive non-hospitalized vs COVID-19 negative or unknown.

- II. The association remained when restricting the analysis to individuals of European ancestry (Supp Table 13), and there was no evidence for heterogeneity of effect between studies ($P_{\text{het}}=0.44$).
- III. Variant rs143334143 was also associated at $P<0.05$ with our two severity phenotypes among cases (Supp Table 14).
- IV. In the latest HGI analysis of COVID-19 hospitalization (B2 phenotype) there is overall support for an association with rs143334143 (see circle in Figure below), but the signal is clearly not as strong as originally reported by Pairo-Castineira et al. There is also evidence for heterogeneity of effects across studies ($P_{\text{het}}=0.0005$), but of course this could arise for a number of reasons, including differences in disease severity between studies.

Based on our independent replication of the original finding and the additional observations above, we conclude that it is highly plausible that rs143334143 tags a true genetic risk factor for COVID-19, although its effect on disease risk was likely overestimated in the Pairo-Castineira et al. study. We do not think that there is enough evidence to justify excluding this variant from our analyses.

8. Furthermore, great caution must be taken when reporting HLA associations – in general, and in trans-ethnic analysis in particular, given the huge risk of cryptic population substructures. I

remain unconvinced on the validity of the CCHCR1 association reported for this reason (unless confirmatory data is provided), as well as the lack of regular clarification you would see for an HLA association (e.g. alignment with classical loci alleles, HLA-C and HLA-B in particular – including the dimorphisms relevant for KIR interactions to which the authors allude). A simple regional association plot would have been useful in clarifying this region in more detail.

Hopefully the response to question #7 above addresses this concern.

9. This reviewer has major problems with the term “target” for genetic findings. Only an extreme minority of GWAS outcomes have proven their efficacy as drug targets, and this way of reporting genetic associations holds a corporate bias in the context of a study within the broader Regeneron scope. I would refer to loci, genes and variants for detected associations, title included.

We thank you for raising these concerns and have removed instances of the word ‘target’ in the context of genetic associations.

10. The referencing to antibody cocktails in the reporting of the GRS holds the same corporate bias and should be deleted (ditto for vaccine), as this is an indirect speculation and lacks direct evidence. The topic of severe Covid-19 cannot afford unfounded clinical suggestions.

We have toned down these references, and are also happy to do it more extensively if the Editors feel like this would be appropriate.

11. The two novel associations arise not from the meta-analysis mentioned under above point 1, but by an expanded meta-analysis using data also from FinnGen, Ellinghaus et al. and the GenOMICC studies. Hits are noted at KAT7 and TMPRSS2 at just about genome-wide significance. Whilst certainly of some interest, TMPRSS2 in particular, I would report the loci with caution in lack of independent validation (are the authors able to bring forward such replication?), and moderate claims related to treatment of patients (e.g. suggesting that inhibition of TMPRSS2 might be protective against Covid-19).

In our revised manuscript, the only novel association that we report is with a rare variant in ACE2. Unfortunately, because this variant is rare, independent well-powered replication was not possible. To have 80% power to replicate the observed association at a $P < 0.05$ we would need approximately 10K cases and 40K controls. We do not have access to independent studies of this scale with data for the rare ACE2 variant.

12. *Could an alternative approach to reporting the analyses seen in this paper have been the two novel cohorts separately, and then the expanded meta-analyses resulting in these two novel findings?*

We do provide results (full summary statistics, top associations, regional association plots, etc) for the two novel cohorts separately through our COVID-19 browser (<https://rgc-covid19.regeneron.com/home>). However, as we discussed above, the approach that provided the greatest statistical power to identify genetic associations with COVID-19 susceptibility and severity was to combine results across all available studies, and so we opted for making that the primary set of analyses reported in this manuscript.

13. *The GRS section and analysis is to this reviewer the most interesting part of the manuscript. How does GRS risk stratification perform in comparison to risk stratification by clinical risk factors?*

In Europeans of the AncestryDNA study, SARS-CoV-2 infected individuals with high clinical risk have 5-fold greater odds of developing severe disease (i.e. respiratory failure or death) when compared to infected individuals with low clinical risk. In comparison, a high GRS (top 10%) was associated with a 1.7-fold increased risk of severe disease. Similar estimates were observed in the UK Biobank study, please see Fig 3.

Reviewer #2:

1. *The authors show in Supplementary Table 2 that their cases and controls had very different levels of comorbidities, such as hypertension, CVD, T2D, CKD, asthma, and COPD. These imbalances of comorbidities, which are known independent risk factors for severe COVID-19, are not included in any of the association models for the GWAS or replication analyses. Therefore, there is potential for substantial confounding in which results measure nothing specific to COVID, but rather measure genetics related to these comorbidities. Sensitivity analyses should be conducted, as well as controlling for comorbidity burden within the regression models.*

This is a very important point that we addressed in the GRS analyses (comorbidities are included as covariates in the logistic regression analyses) but not in the GWAS analyses.

To address the possibility that the six variants identified in previous GWAS and that we validated in our study reflect associations with comorbidities rather than with COVID-19, we performed a GWAS for each comorbidity listed above in the UK Biobank study and extracted results for those

six variants. Results are summarized in the new Supplementary Table 8. We found that only one of the six variants showed a clear association with a clinical risk factor: the MHC variant and risk of asthma, which was expected given the strong association between this region and allergic diseases. Of note, however, the direction of effect on asthma was opposite to that observed for COVID-19. As such, we conclude that it is very unlikely that the association between these six published risk variants for COVID-19 measure genetics related to these comorbidities. Similar findings were observed for the two HGI variants that we highlight in the manuscript and the novel ACE2 risk variant (Supplementary Table 8).

We have added a paragraph to the main text that describes these findings.

2. Another major concern is the use of ‘susceptibility’ as an outcome. Infectious disease requires exposure to the pathogen for an individual to develop the outcome of interest (infection). As the vast majority of controls were likely not exposed during the recruitment time, the comparison for SARS-CoV-2 positive tests to negative tests or no test does not measure true susceptibility. Instead it measures both the behavior of exposure and susceptibility to infection. I suggest changing the terminology to ‘risk of positive test’ or something similar to distinguish this.

We have removed references to “susceptibility” throughout the manuscript, and now use the term “risk of infection” to collectively refer to the five phenotypes that we analyzed that compare infected individuals against those with no record of infection.

3. There were no tests of heterogeneity presented across studies or ancestries as part of the meta-analysis. Given how heterogeneous the effect sizes look in the forest plots, it would be good to present these analyses and interpret why some effect sizes may be wildly different even despite similar sample size.

Done as requested. Heterogeneity test P-values added to Supplementary Tables 7, 9, 12, 13, 14 and 15. There were no associations with clear evidence for heterogeneity.

4. The authors do not currently discuss possible reasons for heterogeneity in case/control status and associated risk factors between studies. It would be beneficial for the manuscript to touch on this, as there are a large number of COVID-19 GWAS being published and no way to distinguish them and their conflicting results. These results need to be contextualized by describing the underlying study population and any benefits/limitations therein.

Done as requested, please see new caveats section at the end of the main text.

5. The GRS and clinical risk factors analyses is important and shows the added benefit of genetics to traditional risk scores. However, there are major differences between the AncestryDNA and UK Biobank results, in which the GRS effects are significantly attenuated in UK Biobank. Additionally, the GRS is further attenuated in risk of hospitalization. The authors should discuss potential reasons for this lack of transferability, such as overfitting for AncestryDNA or differences in healthcare systems and ability to be hospitalized.

Done as requested, please see new caveats section at the end of the main text.

6. *There is no replication for the novel risk variants identified in this study. This should be done either through internal validation, or doing a lookup of the previously published GWAS that were not included in this analysis.*

Please see response to question 11 of reviewer #1.

7. *In general, the manuscript currently reads like a list of results, instead of a description of the results and adequate expansion to explain why the results matter to the reader, as well as provide interpretations of the major points and potential limitations. I understand this is due to the limits of the format as a brief communication, but it would be helpful to rebalance some of the text to help the reader.*

We appreciate the feedback and tried to rebalance the text to provide a more thorough interpretation of results.

8. *The authors briefly touch on the five phenotypes related to disease risk and two phenotypes [related to disease severity]. I would include some sort of flow chart or diagram to understand these within the main text if there is space. It would also be useful to put in some text explaining how these phenotypes differ in a meaningful way.*

Fully agree and now present this information in the new Table 1.

9. I am assuming that “Admixed American” refers to Hispanic/Latino individuals. However, this term is confusing and is not consistent with prior literature or the standard in the field.

Your assumption is correct that individuals who identify as Hispanic/Latino are included in the Admixed American group. We agree that the term, “Admixed American” is probably not the most straightforward term. For better or worse, we followed the continental ancestry labeling from

1000 Genomes Project (<https://www.internationalgenome.org/category/population/>) often used by statistical geneticists.

10. Please put the actual numbers throughout the manuscript instead of <100, especially as some cohorts have this information and some just have <100.

We acknowledge the reviewers request for this information; however, due to privacy concerns around customer data, we are unable to accommodate this request on account of the increased re-identification risk to our customers.

11. It would be helpful to include the effect sizes of the previously published loci and how they compare to your effect sizes in the main text. This would also help to contextualize your results which show differences in magnitude across studies.

Done as requested.

Decision Letter, first revision:

28th May 2021

Dear Manuel,

Your Brief Communication, "Genetic associations with COVID-19 in 756,646 individuals identify a protective variant near ACE2 and patients at high risk of severe disease" has now been seen by the 2 original referees.

You will see from their comments below that while they continue to find your work of interest, some important points are still raised. We are interested in the possibility of publishing your study in Nature Genetics, but would like to consider your response to these concerns in the form of a revised manuscript before we make a final decision on publication.

Briefly, both reviewers think this study has improved in revision. However, each also makes a number of suggestions for further analysis: e.g. Reviewer #1 suggests examining a genome-wide PRS; while Reviewer #2 thinks the sex-specific aspects of COVID-19 risk should be further examined. There are also a number of comments on interpretation and presentation; most importantly, Reviewer #1 questions the independence of the chr3 loci and asks that this be clarified.

We think that these suggestions are not unduly onerous and agree with the reviewers that these

additions would further strengthen the study.

To guide the scope of the revisions, the editors discuss the referee reports in detail within the team, including with the chief editor, with a view to identifying key priorities that should be addressed in revision and sometimes overruling referee requests that are deemed beyond the scope of the current study. We hope that you will find the prioritized set of referee points to be useful when revising your study. Please do not hesitate to get in touch if you would like to discuss these issues further.

We therefore invite you to revise your manuscript taking into account all reviewer and editor comments. Please highlight all changes in the manuscript text file. At this stage we will need you to upload a copy of the manuscript in MS Word .docx or similar editable format.

*2) If you have not done so already please begin to revise your manuscript so that it conforms to our Brief Communication format instructions, available [here](http://www.nature.com/ng/authors/article_types/index.html). Refer also to any guidelines provided in this letter.

[REDACTED]

We hope to receive your revised manuscript within four to eight weeks. If you cannot send it within this time, please let us know.

Please do not hesitate to contact me if you have any questions or would like to discuss these revisions

further.

Sincerely,

Michael Fletcher, PhD
Associate Editor, Nature Genetics

ORCID: 0000-0003-1589-7087

Reviewers' Comments:

Reviewer #1:

Remarks to the Author:

Horowitz et al. have re-written and re-focused the manuscript extensively in response to the reviewers comments. The revised manuscript appears considerably cleaner with regards to results presentation and robustness. I am happy with the inclusion of the MHC signal, given the current discussion. I am also happy for the ACE2 rare variant signal to be included, despite the lack of validation. That said, I do have a few remaining comments for consideration by the authors:

1) I have problems with the claims of two independent reports for chromosome 3 (still) – LZTFL1 and SLC6A20. The core haplotype (as extensively described by Zeberg and Pääbo (<https://www.nature.com/articles/s41586-020-2818-3>) is inherited as a whole, and includes both LZTFL1 and SLC6A20 ($r^2 > 0.98$ in their Nature paper). I don't question the appearance of the signals at apparent independency (and opposite effects) during statistical analysis, but this is likely a function of cohort-"substructures" resulting from the variable phenotype definitions, and I would encourage the authors in finding a different way of presenting this association (representing the same, underlying, and correlated genetic risk) – also impacting on the assembly of the "six gene panel" carried further into the analysis (leaving this with 5 genes, re. below).

2) I would shorten the "analogies" section on page 8 with the various comparisons versus the C19-HGI flagship paper. Maybe a single sentence notion would suffice.

3) There are some claims of "associations" without significant p-values (e.g. the South Asian GRS on p11 – a very interesting GRS indeed, given the high carrier rate of the chromosome 3 variant in this

ethnic group – and current high burden of severe Covid-19 in India) and some notions of “trends” (p12 on the GRS). Please reporting findings plainly/without “flavor”, with or without statistical significance.

4) The GRS section is still what holds the strongest novelty and interest to this reviewer. I am happy for the expansion on the demarcation versus clinical risk factors, and would somehow encourage authors to briefly reference also the other C19-HGI paper currently in processing (<https://www.medrxiv.org/content/10.1101/2021.03.07.21252875v1>) similar to the referencing of the “main” C19-HGI paper, their results are complementary and not competitive. Given the awareness of genetic risk over and above the 6 (or 5...) gene panel, which is now even acknowledged by the authors re. pt. 2 above (and their aligning of own results with the C19-HGI), I think most readers (this reviewer included) would have expected to see a “proper” GW polygenic risk score (PRS) performance comparison with their 6/5 “gene panel”, to assess whether the limitation of genetic risk assessment to these 6/5 signals is valid, or a broader PRS would be more appropriate.

Reviewer #2:

Remarks to the Author:

The reviewers have addressed many of my original comments. I still have a few outstanding suggestions that would improve the impact of the manuscript as well as clarify some conclusions. Overall, the manuscript would benefit with a restructuring to really emphasize the portions that are unique to this manuscript. The focus on the chromosome X rare variant for protection is great and I would also expand on the GRS since this is unique to this manuscript and study. I would condense the “replication” part of it, given that seems to be a large focus of many other manuscripts using the same samples.

1. For the GRS analysis, it would be useful to quantify the added benefit above and beyond traditional risk factors. For example, what proportion of people who are “low risk” from clinical risk factors would then be determined to be “high risk” based on the GRS? Right now this section seems superficial and doesn’t make the comparison which is the most useful, i.e. what does this really add on top of known risk factors. It also doesn’t give the usual prediction metrics, such as AUC. The first paragraph cites that genetic risk gives “greater precision” but this isn’t supported by the specific analyses that are done in the actual manuscript.
2. It is unclear if the “C2” analysis variants, which represent infection vs no known infection, were also identified in the severity phenotypes from the main text. I would like to see more of an explanation of the value of this section.
3. I would like to see the role of a chromosome X in possibly explaining sex-specific differences explored. For example, what proportion of variance is explained by that variant in the female vs male subsets? The effect seems stronger in males (even though heterogeneity P-value is >0.05).
4. The manuscript focuses on results in European-descent populations, which I understand given the very different sample sizes. However, it would be nice to know if the same trends hold comparing the predictive accuracy of clinical risk factors vs GRS for all ancestries. Right now the authors only present how the effect sizes for high-risk GRS differ by ancestry, but it would be useful to have the same comparisons as EUR groups.

Author Rebuttal, first revision:

Reviewer #1

1. I have problems with the claims of two independent reports for chromosome 3 (still) – LZTFL1 and SLC6A20. The core haplotype (as extensively described by Zeberg and Pääbo is inherited as a whole, and includes both LZTFL1 and SLCA620 ($r^2 > 0.98$ in their Nature paper). I don't question the appearance of the signals at apparent independency (and opposite effects) during statistical analysis, but this is likely a function of cohort-“substructures” resulting from the variable phenotype definitions, and I would encourage the authors in finding a different way of presenting this association (representing the same, underlying, and correlated genetic risk) – also impacting on the assembly of the “six gene panel” carried further into the analysis (leaving this with 5 genes, re. below).

Thank you for pointing us to the Zeberg and Paabo manuscript, which we were not aware of. In Supplementary Table 5 we list two published risk variants in the chromosome 3 locus: rs73064425 in LZTFL1, and rs2531743 in SLC6A20. We list these two variants because they are in low LD with each other ($r^2=0.02$ in Europeans), which indicates that these two variants are not inherited as a whole.

Zeberg and Paabo describe a “core haplotype” that contains about 10 variants in high LD with each other ($r^2>0.8$) and they also highlight many other variants more modestly correlated with this haplotype (specifically with $r^2>0.1$; see their Figure 1). But this haplotype block (indexed by rs35044562) is in high LD with the LZTFL1 variant rs73064425 ($r^2=0.99$) but not the SLC6A20 variant rs2531743 ($r^2=0.02$). So the “core haplotype” the authors describe at chromosome 3 for the C2 phenotype is not related to the SLC6A20 variant that Shelton et al (PMID 33888907) reported for the A2 phenotype

For these reasons, we believe it is appropriate to consider these two variants separately in our manuscript.

2. I would shorten the “analogies” section on page 8 with the various comparisons versus the C19-HGI flagship paper. Maybe a single sentence notion would suffice.

Done as requested.

3. There are some claims of “associations” without significant p-values (e.g. the South Asian GRS on p11 – a very interesting GRS indeed, given the high carrier rate of the chromosome 3 variant in this ethnic group – and current high burden of severe Covid-19 in India) and some notions of “trends” (p12 on the GRS). Please reporting findings plainly/without “flavor”, with or without statistical significance.

Done as requested.

4. The GRS section is still what holds the strongest novelty and interest to this reviewer. I am happy for the expansion on the demarcation versus clinical risk factors, and would somehow encourage authors to briefly reference also the other C19-HGI paper currently in processing (<https://www.medrxiv.org/content/10.1101/2021.03.07.21252875v1>) similar to the referencing of

the “main” C19-HGI paper, their results are complementary and not competitive. Given the awareness of genetic risk over and above the 6 (or 5...) gene panel, which is now even acknowledged by the authors re. pt. 2 above (and their aligning of own results with the C19-HGI), I think most readers (this reviewer included) would have expected to see a “proper” GW polygenic risk score (PRS) performance comparison with their 6/5 “gene panel”, to assess whether the limitation of genetic risk assessment to these 6/5 signals is valid, or a broader PRS would be more appropriate.

As suggested, we have now compared the performance of the 6-SNP genetic risk score (GRS) with a full genome-wide polygenic risk score, using various approaches (pruning and thresholding and LDpred). We found that genome-wide scores had similar (not better) associations with risk of hospitalization and disease severity when compared to the 6-SNP GRS. Please see new Supplementary Figure 5.

Reviewer #2

1. I would condense the “replication” part of it, given that seems to be a large focus of many other manuscripts using the same samples.

Done as requested.

2. For the GRS analysis, it would be useful to quantify the added benefit above and beyond traditional risk factors. For example, what proportion of people who are “low risk” from clinical risk factors would then be determined to be “high risk” based on the GRS? Right now this section seems superficial and doesn’t make the comparison which is the most useful, i.e. what does this really add on top of known risk factors. It also doesn’t give the usual prediction metrics, such as AUC. The first paragraph cites that genetic risk gives “greater precision” but this isn’t supported by the specific analyses that are done in the actual manuscript.

We agree that we were not focusing on the most relevant comparison, as the reviewer alludes to, and have now rectified this. We have also included a new figure with AUC estimates (see Supplementary Figure 5).

2. It is unclear if the “C2” analysis variants, which represent infection vs no known infection, were also identified in the severity phenotypes from the main text. I would like to see more of an explanation of the value of this section.

We have streamlined this section and tried to clarify the value and novelty. Specifically, our goal was to identify a set of variants associated with severity, which we now state explicitly. We first attempted to replicate variants identified in previous GWAS and to determine which of these were specifically associated with severity (identifying 4 variants). We next explored which variants identified in the HGI “C2” analysis (infected vs no record of infection) were also associated with severity in our data (e.g. infected but mild vs infected but severe), identifying a further two variants of interest.

3. I would like to see the role of a chromosome X in possibly explaining sex-specific differences explored. For example, what proportion of variance is explained by that variant in the female vs male subsets? The effect seems stronger in males (even though heterogeneity P-value is >0.05).

We now note the variance explained by the variant in males and females (page 4, paragraph 3). Variance explained is larger in males, both because the estimated additive genetic effect is slightly larger in males (p-value = 0.009, in our current updated dataset) and because the standard biometric model predicts a larger contribution of X-chromosome to variance in males, after accounting for dosage compensation (e.g. for a variant with frequency p and additive effect a , estimated variance would be $2p(1-p)a^2$ in females and $4p(1-p)a^2$ in males. However, because the variant is so rare, it cannot explain the observed differences in risk between males and females.

4. The manuscript focuses on results in European-descent populations, which I understand given the very different sample sizes. However, it would be nice to know if the same trends hold comparing the predictive accuracy of clinical risk factors vs GRS for all ancestries. Right now the authors only present how the effect sizes for high-risk GRS differ by ancestry, but it would be useful to have the same comparisons as EUR groups.

Given the limited sample sizes available, we should be cautious in interpreting results in non-European ancestry individuals. That said, we now include additional detail in our discussion of **Supplementary Table 18** and the value of the GRS appears remarkably consistent across ancestries. We have also performed the GRS analysis stratified by clinical risk factors in individuals of Admixed American ancestry, with very consistent results (see new Supplementary Figure 4). Sample size was too small to perform this analysis in the other non-European Ancestries.

Decision Letter, second revision:

30th Jun 2021

Dear Manuel,

Your Brief Communication, "Genome-wide analysis in 756,646 individuals provides first genetic evidence that ACE2 expression influences COVID-19 risk and yields genetic risk scores predictive of severe disease" has now been seen by the original 2 referees.

You will see from their comments below that while they continue to find your work of interest, there are still important issues remaining to be resolved. We are interested in the possibility of publishing your study in Nature Genetics, but would like to consider your response to these concerns in the form of a revised manuscript before we make a final decision on publication.

Briefly, both reviewers seem largely satisfied with the analysis as performed. However, both also have issues with the presentational aspects of the manuscript.

Reviewer #1 is happy with the responses to the previous round of review, but suggests that the details of the LZTFL1/SLC6A20 locus should be included in the manuscript (not only the rebuttal). They also point out missing citations of past literature.

Reviewer #2, on the other hand, thinks that the manuscript text is lacking in detail in many areas. They suggest that these details could be substantially, and fruitfully for the reader, expanded upon. They also point out that the use of "Admixed American" is incorrect and should be replaced with an accurate, up-to-date label of sample ancestry.

Overall, we think that these are important, but addressable, issues. The major comments in both reviews touch upon issues of presentation; we suggest that an expansion of the main text would provide this requested detail. Reformatting your article from a Brief Communication to a Letter would be in line with such an expansion of the text, and we'd recommend that.

To guide the scope of the revisions, the editors discuss the referee reports in detail within the team, including with the chief editor, with a view to identifying key priorities that should be addressed in revision and sometimes overruling referee requests that are deemed beyond the scope of the current study. We hope that you will find the prioritized set of referee points to be useful when revising your study. Please do not hesitate to get in touch if you would like to discuss these issues further.

We therefore invite you to revise your manuscript taking into account all reviewer and editor comments. Please highlight all changes in the manuscript text file. At this stage we will need you to upload a copy of the manuscript in MS Word .docx or similar editable format.

We are committed to providing a fair and constructive peer-review process. Do not hesitate to contact us if there are specific requests from the reviewers that you believe are technically impossible or

unlikely to yield a meaningful outcome.

*2) If you have not done so already please begin to revise your manuscript so that it conforms to our Brief Communication format instructions, available [here](http://www.nature.com/ng/authors/article_types/index.html). Refer also to any guidelines provided in this letter.

[REDACTED]

We hope to receive your revised manuscript within four to eight weeks. If you cannot send it within this time, please let us know.

Sincerely,

Michael Fletcher, PhD
Associate Editor, Nature Genetics

ORCID: 0000-0003-1589-7087

Reviewers' Comments:

Reviewer #1:

Remarks to the Author:

Somehow the tracing of changes made were not straightforward due to lack of markup of changes made in the version received by this reviewer (possibly a technical issue), but re-reading the entire manuscript, the changes referred to in the rebuttal appear acceptable - with one exception. I (still) think the discussion related to rs73064425 in LZTFL1 versus rs2531743 in SLC6A20 warrant mentioning in the text, very well according to directions given by the rebuttal. Given the "cohort-dependency" of the apparent independent associations (LZTFL1 for severity, SLC6A20 for susceptibility), I retain some residual concern that underlying bias resulting from "phenotype substructures" may be involved, but with an appropriate discussion I can live with the choice of presentation chosen by the authors.

It appears somehow odd that key landmark papers of the field are not cited, at minimum the first GWAS for Covid-19 which was published in NEJM in June last year (PMID: 32558485) should be mentioned, since the two loci reported overlap with the main findings of the present work. Presumably the authors are aware of this literature and should acknowledge its existence according to good scientific practice.

Reviewer #2:

Remarks to the Author:

The authors have revised their manuscript and added additional analyses. However, most of these analyses are in the supplement and are not mentioned or described in the main text. Additionally, the language throughout the manuscript is imprecise and vague, making it difficult to understand without digging into the figure and table legends quite often. My specific comments are below.

- While the authors claim to have addressed the previous reviews in their response to reviewers, many of the additional analyses are only found in the supplement and not in the main text. It is not adequate to stuff this information in the supplement without at least some information in the main text describing the results.

- I had previously suggested that the authors investigate the added benefit of a genetic risk score to a clinical risk score. In response they cited supplementary figure 5. While this figure does include AUC estimates, it does not address the question of clinical versus genetic risk score. For this I think they

actually meant supplementary figure 6 and this does not address the question, nor is it described in the main text.

- Throughout the manuscript, the authors are extremely vague as to what specific phenotype they are talking about. They are not interchangeable (at least they shouldn't) and precision with language is warranted. The only place to gain clarity for what the authors are actually comparing is in Table 1. Many associations are not specific, including the main first part of the article, requiring readers to dig into the figure legends to find out what comparisons are being made.
- There are many times throughout the manuscript that the authors make a statement without actually telling us the numbers. Readers should not have to dig through supplementary tables and figures in order to find out how much is "nominally" significant or "consistent" effects. This includes the additional analyses they have added since the last review, such as the extended risk score analyses (page 10, lines 210-211).
- It is unclear why the authors chose to select SNPs that were found for both "infection" and severity outcomes. This should be justified in the manuscript, especially as many other groups have found distinct genetic signatures between risk of infection and severe disease once infected.
- On page 6, lines 124-125, the authors note that two of the 6 variants are moderately associated with clinical risk factors but that they are unlikely to explain their association with COVID-19. What happens to the effect size between these variants and the COVID-19 outcomes when you adjust for these comorbidities specifically? That would be the way to check this assumption.
- Figure 2 shows that only the top 5% of the GRS is significantly associated with hospitalization and severe disease. Figure 3 uses the top 10%. Why are they different?
- The authors have previously justified the use of "Admixed American" to denote individuals who are Hispanic/Latino because 1000Genomes uses this abbreviation. However, the 1000Genomes denotes AMR as "American" and this is not the standard in the field when describing populations and their health as it is imprecise. I suggest the authors change this label to be consistent with who they are actually describing in accordance with more up-to-date standards (such as <https://genomebiology.biomedcentral.com/articles/10.1186/s13059-018-1396-2>). This is especially relevant as these groups were disproportionately impacted by COVID. The information is not useful to them if the labels are imprecise and no one knows what you are talking about.

Author Rebuttal, second revision:

Reviewer #1

1. I (still) think the discussion related to rs73064425 in LZTFL1 versus rs2531743 in SLC6A20 warrant mentioning in the text, very well according to directions given by the rebuttal. Given the "cohort-dependency" of the apparent independent associations (LZTFL1 for severity, SLC6A20 for susceptibility), I retain some residual concern that underlying bias resulting from "phenotype substructures" may be involved, but with an appropriate discussion I can live with the choice of presentation chosen by the authors.

Done as requested. We added the following text on page 7 (lines 132 to 137):

“The variants in LZTFL1 and SLC6A20 are located 63 Kb apart at the 3p21.31 locus first reported by Ellinghaus et al.⁶, which contains a core risk haplotype that includes 13 variants in high LD

with each other¹⁷. However, in individuals of European ancestry, this haplotype block (indexed by rs35044562) is in high LD with the *LZTFL1* variant rs73064425 ($r^2=0.99$) but not the *SLC6A20* variant rs2531743 ($r^2=0.02$), indicating that these two signals – for severe disease and risk of infection, respectively – are likely independent.”

Importantly, only one of these two variants is included in our 6-SNP GRS (rs73064425 in *LZTFL1*), and so this issue of independence between the *LZTFL1* and *SLC6A20* variants does not in any way affect our GRS analyses.

2. It appears somehow odd that key landmark papers of the field are not cited, at minimum the first GWAS for Covid-19 which was published in NEJM in June last year (PMID: 32558485) should be mentioned, since the two loci reported overlap with the main findings of the present work. Presumably the authors are aware of this literature and should acknowledge its existence according to good scientific practice.

Thank you for pointing this out. We did cite all previously published GWAS in Supp Table 5 (PubMed IDs provided in column C), but some of these studies were not directly mentioned in the main text, which explains why they do not appear in the main references section. However, we now directly mention in the main text the NEJM paper the reviewer alludes to, as well as the studies by GenOMICC (Pairo-Castineira et al.) and 23andMe (Shelton et al.). Please see new text and citations added to page 7 (and quoted in response to comment #1 above) as well as the citations added to page 4.

Reviewer #2

1. While the authors claim to have addressed the previous reviews in their response to reviewers, many of the additional analyses are only found in the supplement and not in the main text. It is not adequate to stuff this information in the supplement without at least some information in the main text describing the results.

Yes, we agree that we relied too heavily on the supplement and could have done a better job at providing a brief summary of the additional analyses performed in the main text. To address this issue, we identified the key set of results that were in the supplement, but not adequately explained in the main text, and made the following changes:

- Results between *ACE2* variant and risk of hospitalization among cases are now summarized on page 5.
- Details around the six variants that replicated in our study are now summarized in page 6. It should now be easier for readers to directly compare our results with those from previous studies using information provided in this section. We had included this paragraph in our first submission, but subsequently removed it because of space constraints.

- We now briefly expand upon the association between two variants (in the MHC and *ABO*) and clinical risk factors on page 7.
- We have considerably expanded the description of the genetic risk score analysis on pages 11 and 12. We fully acknowledge that we had summarized these analyses too extensively in the previous submission (this also addresses point #4 below).

2. I had previously suggested that the authors investigate the added benefit of a genetic risk score to a clinical risk score. In response they cited supplementary figure 5. While this figure does include AUC estimates, it does not address the question of clinical versus genetic risk score. For this I think they actually meant supplementary figure 6 and this does not address the question, nor is it described in the main text.

We apologize for not having addressed the question of clinical versus genetic risk score explicitly in our previous submission. This is important and we now include a new section on pages 12 to 13 that tackles this question head on. Overall, we found that the GRS modestly improved risk prediction (specifically the AUC for disease severity improved by 0.8% and 0.5%, respectively in the AncestryDNA and UKB studies) when added to a model that included non-genetic risk factors, such as age, sex and clinical risk factors. This magnitude of improvement in the AUC was comparable to that observed with some clinical risk factors individually, such as cardiovascular disease (0.6% and 0.5%, respectively in AncestryDNA and UKB). These results are also illustrated in the new Figure 4.

3. Throughout the manuscript, the authors are extremely vague as to what specific phenotype they are talking about. They are not interchangeable (at least they shouldn't) and precision with language is warranted. The only place to gain clarity for what the authors are actually comparing is in Table 1. Many associations are not specific, including the main first part of the article, requiring readers to dig into the figure legends to find out what comparisons are being made.

Thank you for pointing that it wasn't clear which specific phenotypes we were discussing. This is quite critical, as the phenotypes tested are not at all interchangeable. We have now added more detail around which phenotypes were analyzed throughout the text, including in pages 4, 5, 6, 8 and 10.

4. There are many times throughout the manuscript that the authors make a statement without actually telling us the numbers. Readers should not have to dig through supplementary tables and figures in order to find out how much is "nominally" significant or "consistent" effects. This includes the additional analyses they have added since the last review, such as the extended risk score analyses (page 10, lines 210-211).

We systematically went through the manuscript to identify such situations, and updated the text with more details around specific associations accordingly (including P-values).

5. *It is unclear why the authors chose to select SNPs that were found for both “infection” and severity outcomes. This should be justified in the manuscript, especially as many other groups have found distinct genetic signatures between risk of infection and severe disease once infected.*

This is now clarified on page 10:

“In these initial analyses, we did not consider variants that were not associated with disease severity among cases (e.g. in/near SLC6A20 and ABO) because these variants were not expected to be informative to assess risk of severe disease once infected with SARS-CoV-2. Consistent with this expectation, as we describe later, sensitivity analyses show that including such variants in the GRS did not improve associations with severe disease.”

6. *On page 6, lines 124-125, the authors note that two of the 6 variants are moderately associated with clinical risk factors but that they are unlikely to explain their association with COVID-19. What happens to the effect size between these variants and the COVID-19 outcomes when you adjust for these comorbidities specifically? That would be the way to check this assumption.*

The two variants in question are rs143334143 in the MHC and rs879055593 in *ABO*, both reported in previously published GWAS of COVID-19. First, we would like to point out that only the MHC variant is included in our 6-SNP GRS. Second, our GRS analyses adjust for clinical risk factors, and so even if the association between the MHC variant and COVID-19 was explained by clinical risk factors, that would have been adequately controlled for in our analyses. Therefore, we do not believe that the issue raised by the reviewer impacts our results.

Nonetheless, to fully address the reviewer’s concern, we estimated the effect size of both the MHC and *ABO* variants in the UKB study after adjusting for the associated clinical risk factors. A summary of these analyses is provided below. Overall, the effect sizes for both variants remained essentially unchanged after adjusting for clinical risk factors. We have included this information in the new Supplementary Table 14.

1. rs143334143 in the MHC

- a. Reported to be associated with COVID-19 by Pairo-Castineira et al. (PMID 33307546)
- b. Replicated in our study in the analysis of COVID-19 positive vs. COVID-19 negative (OR=1.06, $P=2 \times 10^{-4}$)
- c. Found in our study to be associated with two clinical risk factors for COVID-19:
 - i. Asthma: OR=0.91, $P=7 \times 10^{-9}$. This is the opposite direction of effect as observed for COVID-19, so the association with asthma cannot explain the association with COVID-19.
 - ii. Type-2 diabetes: OR=1.08, $P=2 \times 10^{-5}$.

- d. Effect size for this variant in the analysis of COVID-19 positive vs. COVID-19 negative in the UKB study:
 - i. Without controlling for asthma or type-2 diabetes: OR=1.08 (1.03, 1.14), P=0.004.
 - ii. Controlling for asthma and type-2 diabetes: OR=1.07 (1.00, 1.14), P=0.044.
2. rs879055593 in ABO
- a. First reported to be associated with COVID-19 by Ellinghaus et al. (PMID 32558485).
 - b. Replicated in our study in the analysis of COVID-19 positive vs. COVID-19 negative or unknown (OR=1.10, P=7x10⁻³⁴)
 - c. Found in our study to be associated with two clinical risk factors for COVID-19:
 - i. Kidney disease: OR=0.96, P=10⁻⁴. This is the opposite direction of effect as observed for COVID-19, so the association with kidney disease cannot explain the association with COVID-19.
 - ii. Type-2 diabetes: OR=1.04, P=10⁻⁴. This effect size is considerably smaller than that observed for COVID-19, so again the association with type-2 diabetes cannot explain the association with COVID-19.
 - d. Effect size for this variant in the analysis of COVID-19 positive vs. COVID-19 negative or unknown in the UKB study:
 - i. Without controlling for kidney disease or type-2 diabetes: OR=1.09 (1.06, 1.12), P=2x10⁻¹⁰.
 - ii. Controlling for kidney disease and type-2 diabetes: OR=1.08 (1.04, 1.11), P=8.56x10⁻⁶.

7. *Figure 2 shows that only the top 5% of the GRS is significantly associated with hospitalization and severe disease. Figure 3 uses the top 10%. Why are they different?*

First, we would like to point out that Figure 2 actually shows that all GRS cut-offs tested have a significant association with hospitalization and severe disease. And, in fact, the association with risk of severe disease for the top 10% was more significant than for the top 5% (P=7x10⁻¹⁰ compared to P=2x10⁻⁷), although numerically the effect sizes are similar (1.58 [95% CI 1.36 to 1.82] compared to 1.66 [95% CI 1.37 to 2.01]).

Figure 3 shows results for the same analysis but now after stratifying COVID-19 cases into two groups: those with low and high clinical risk. For this stratified analysis we did not define high genetic risk based on the top 5% of the GRS because the resulting number of individuals with a high GRS would be too small for analysis.

8. The authors have previously justified the use of “Admixed American” to denote individuals who are Hispanic/Latino because 1000Genomes uses this abbreviation. However, the 1000Genomes denotes AMR as “American” and this is not the standard in the field when describing populations and their health as it is imprecise. I suggest the authors change this label to be consistent with who they are actually describing in accordance with more up-to-date standards (such as <https://genomebiology.biomedcentral.com/articles/10.1186/s13059-018-1396-2>). This is especially relevant as these groups were disproportionately impacted by COVID. The information is not useful to them if the labels are imprecise and no one knows what you are talking about.

Thank you for insisting on this point; we agree that imprecise labels impede the dissemination of information. We were following the nomenclature used by the 1000 Genomes project, but clearly that is not satisfactory and out-of-date, as pointed out by the Morales et al. (2018) paper the reviewer cites above (PMID 29448949). Table S4 in Morales et al. proposes that the populations labeled as “AMR” in the 1000 Genomes Project should be labeled instead “Hispanic or Latin American”. As such, we have made this change throughout the manuscript as it best reflects the ancestries of those individuals in our study.

Decision Letter, third revision:

9th Aug 2021

Dear Manuel,

Your Brief Communication, "Genome-wide analysis in 756,646 individuals provides first genetic evidence that ACE2 expression influences COVID-19 risk and yields genetic risk scores associated with severe disease" has now been seen by one of the original referees.

You will see from their comments below that while they continue to find your work of interest and think it has improved during the past round of revision, they do still have a number of issues. We are interested in the possibility of publishing your study in Nature Genetics, but would like to consider your response to these concerns in the form of a revised manuscript before we make a final decision on publication.

Briefly, Reviewer #2 thinks that the analytical framework - nesting phenotypes to identify variants - needs to be more clearly explained, to help better contextualise the results. They also have a number of minor comments regarding the text.

To guide the scope of the revisions, the editors discuss the referee reports in detail within the team, including with the chief editor, with a view to identifying key priorities that should be addressed in revision and sometimes overruling referee requests that are deemed beyond the scope of the current study. We hope that you will find the prioritized set of referee points to be useful when revising your study. Please do not hesitate to get in touch if you would like to discuss these issues further.

We therefore invite you to revise your manuscript taking into account all reviewer and editor comments. Please highlight all changes in the manuscript text file. At this stage we will need you to upload a copy of the manuscript in MS Word .docx or similar editable format.

*2) If you have not done so already please begin to revise your manuscript so that it conforms to our Brief Communication format instructions, available [here](http://www.nature.com/ng/authors/article_types/index.html). Refer also to any guidelines provided in this letter.

[REDACTED]

We hope to receive your revised manuscript within four to eight weeks. If you cannot send it within this time, please let us know.

Nature Genetics is committed to improving transparency in authorship. As part of our efforts in this direction, we are now requesting that all authors identified as 'corresponding author' on published papers create and link their Open Researcher and Contributor Identifier (ORCID) with their account on the Manuscript Tracking System (MTS), prior to acceptance. ORCID helps the scientific community achieve unambiguous attribution of all scholarly contributions. You can create and link your ORCID

from the home page of the MTS by clicking on 'Modify my Springer Nature account'. For more information please visit www.springernature.com/orcid.

Sincerely,

Michael Fletcher, PhD
Associate Editor, Nature Genetics

ORCID: 0000-0003-1589-7087

Reviewers' Comments:

Reviewer #2:

Remarks to the Author:

The authors have addressed the majority of my concerns. I appreciate them addressing even the points that stemmed from my misunderstanding/misreading of figures. I only have one remaining overarching point to bring up.

My only remaining concern is the framing of nested phenotypes around filtering variants. I had previously brought this up in my concern #5 but I think the authors may have misunderstood my point. The authors use infection vs no report of infection as a first step to prioritize variants. They then subset those with variants that are associated with severity given infection. Given these analyses and others have shown that there are several signals that are unique to each of these phenotypes, it's unclear why there is this framing as severity being nested within reported infection. The authors should expand as to why this nesting was used to prioritize variants, especially when the authors expand their GRS to variants only associated with risk of reported infection in that there is no additional predictive ability for severe disease (page 11, lines 238-251), which is expected given they are different phenotypes with different etiologies. Including variants specific to severe disease would make more sense to expand a GRS for severe disease.

Beyond this point there are only a few other extremely minor points that are more to help the readability of the manuscript.

- Page 7, line 145 starts with "To evaluate whether genetics could be used to predict severe disease", but the next few paragraphs have nothing to do with prediction and more with trying to identify genes that may be related to outcomes. The prediction goes back to the original 6 variants. I'm not sure if this is a typo or if I'm just missing the jump between the potential effector genes and the GRS on page 9.

- The first paragraph of page 10 has sample sizes, which is immensely helpful in interpreting the results. Subsequent paragraphs on page 10 and 11 don't have the sample size. It would be a great help to have those in order to better interpret differences in effect sizes and p-values in different subgroups.

- I don't think you actually need to have the sensitivity analyses for the GRS including variants only from reported infection risk in the main text. This is not surprising and would save you some real estate if you are pressed for word count. A sentence or two would suffice as this is not surprising given those added variants are for a different phenotype.

Author Rebuttal, second revision:

Reviewer #2

1. My only remaining concern is the framing of nested phenotypes around filtering variants. I had previously brought this up in my concern #5 but I think the authors may have misunderstood my point. The authors use infection vs no report of infection as a first step to prioritize variants. They then subset those with variants that are associated with severity given infection. Given these analyses and others have shown that there are several signals that are unique to each of these phenotypes, it's unclear why there is this framing as severity being nested within reported infection. The authors should expand as to why this nesting was used to prioritize variants, especially when the authors expand their GRS to variants only associated with risk of reported infection in that there is no additional predictive ability for severe disease (page 11, lines 238-251), which is expected given they are different phenotypes with different etiologies. Including variants specific to severe disease would make more sense to expand a GRS for severe disease.

Thank you once again for insisting on additional clarification for an important section of the manuscript. If it was not clear to the reviewer why we used associations with both risk of infection and disease severity to select variants for the GRS, then it is very likely that some readers would also have the same problem.

We have now edited the relevant paragraph to provide a clear justification for the process used to select variants for the GRS. This section (pages 9 and 10) now reads (new sentences are in red):

“Next, we proceeded to evaluate if common genetic variants can help identify individuals at high risk of severe COVID-19 once infected with SARS-CoV-2. To this end, we created a weighted genetic risk score (GRS) for individuals with a record of SARS-CoV-2 infection and then compared the risk of hospitalization (hospitalized cases vs. non-hospitalized cases) and severe disease (severe cases vs. non-hospitalized cases) between those with a high GRS and all other cases, after adjusting for established risk factors. We considered different approaches to select variants for inclusion in the GRS. First, we reasoned that variants most informative for prediction of severe disease were those associated with worse disease outcomes among infected individuals, and so this was the approach taken for our primary GRS analysis. Of all published genetic risk factors for COVID-19, only one variant was associated with worse outcomes among infected individuals at $P < 5 \times 10^{-8}$ in our analysis (rs73064425 in *LZTFL1*), but this likely reflects low power due to the small number of patients with severe illness that were available for analysis. To address

this limitation, we also included in the GRS five additional variants (in/near MHC, *DPP9*, *IFNAR2*, *RPL24* and *FOXP4*) that (i) had an association with risk of infection at $P < 5 \times 10^{-8}$ in published GWAS or by the HGI; and (ii) we showed above associate with worse disease outcomes among infected individuals, albeit at the suggestive level with current sample sizes. The combination of a genome-wide significant association with risk of infection in previous GWAS and a suggestive association with worse outcomes among infected individuals in the current analysis minimizes the chance that these loci represent false-positive associations for disease severity. Of note, we did not include in the GRS five additional variants discovered by the HGI for risk of hospitalization or severe disease (**Supplementary Table 16**) because the HGI analysis for those two phenotypes is not statistically independent from our analysis of disease outcomes among infected individuals (due to sample overlap). To calculate the GRS, the weights used for each of the six variants corresponded to the effect size (log of the odds ratio) reported in previous GWAS.”

2. Page 7, line 145 starts with "To evaluate whether genetics could be used to predict severe disease", but the next few paragraphs have nothing to do with prediction and more with trying to identify genes that may be related to outcomes. The prediction goes back to the original 6 variants. I'm not sure if this is a typo or if I'm just missing the jump between the potential effector genes and the GRS on page 9.

We agree with the reviewer. The start to this sentence was not entirely appropriate, which we have now fixed. Specifically, we replaced “To evaluate whether genetics could be used to predict severe disease, we first investigated which” with “We then investigated which”.

3. The first paragraph of page 10 has sample sizes, which is immensely helpful in interpreting the results. Subsequent paragraphs on page 10 and 11 don't have the sample size. It would be a great help to have those in order to better interpret differences in effect sizes and p-values in different subgroups.

Done as requested.

4. I don't think you actually need to have the sensitivity analyses for the GRS including variants only from reported infection risk in the main text. This is not surprising and would save you some real estate if you are pressed for word count. A sentence or two would suffice as this is not surprising given those added variants are for a different phenotype.

We would prefer to keep this section as is. Some readers may appreciate the extra detail.

Decision Letter, fourth revision:

Our ref: NG-BC56958R3

31st Aug 2021

Dear Manuel,

Thank you for submitting your revised manuscript "Genome-wide analysis in 756,646 individuals provides first genetic evidence that ACE2 expression influences COVID-19 risk and yields genetic risk scores associated with severe disease" (NG-BC56958R3). It has now been seen by the original referees and their comments are below. The reviewers find that the paper has improved in revision, and therefore we'll be happy in principle to publish it in Nature Genetics, pending minor revisions to satisfy the referees' final requests and to comply with our editorial and formatting guidelines.

Sincerely,

Michael Fletcher, PhD
Associate Editor, Nature Genetics

ORCID: 0000-0003-1589-7087

Reviewer #2 (Remarks to the Author):

The authors have addressed my remaining concerns. Thank you their attention to detail and willingness to address them in prior reviews.

Final Decision Letter:

In reply please quote: NG-A56958R4 Ferreira

17th Dec 2021

Dear Manuel,

I am delighted to say that your manuscript "Genome-wide analysis provides genetic evidence that ACE2 expression influences COVID-19 risk and yields risk scores associated with severe disease" has been accepted for publication in an upcoming issue of Nature Genetics.

Your paper will be published online after we receive your corrections and will appear in print in the next available issue. You can find out your date of online publication by contacting the Nature Press Office (press@nature.com) after sending your e-proof corrections. Now is the time to inform your Public Relations or Press Office about your paper, as they might be interested in promoting its publication. This will allow them time to prepare an accurate and satisfactory press release. Include your manuscript tracking number (NG-A56958R4) and the name of the journal, which they will need when they contact our Press Office.

Please note that *Nature Genetics* is a Transformative Journal (TJ). Authors may publish their research with us through the traditional subscription access route or make their paper immediately open access through payment of an article-processing charge (APC). Authors will not be required to make a final decision about access to their article until it has been accepted. [Find out more about Transformative Journals](https://www.springernature.com/gp/open-research/transformative-journals)

Authors may need to take specific actions to achieve a

<https://www.springernature.com/gp/open-research/funding/policy-compliance-faqs> with funder and institutional open access mandates. For submissions from January 2021, if your research is supported by a funder that requires immediate open access (e.g. according to [Plan S principles](https://www.springernature.com/gp/open-research/plan-s-compliance)) then you should select the gold OA route, and we will direct you to the compliant route where possible. For authors selecting the subscription publication route our standard licensing terms will need to be accepted, including our [self-archiving policies](https://www.springernature.com/gp/open-research/policies/journal-policies). Those standard licensing terms will supersede any other terms that the author or any third party may assert apply to any version of the manuscript.

Please note that Nature Research offers an immediate open access option only for papers that were first submitted after 1 January, 2021.

If you have not already done so, we invite you to upload the step-by-step protocols used in this manuscript to the Protocols Exchange, part of our on-line web resource, natureprotocols.com. If you complete the upload by the time you receive your manuscript proofs, we can insert links in your article that lead directly to the protocol details. Your protocol will be made freely available upon publication of your paper. By participating in natureprotocols.com, you are enabling researchers to more readily reproduce or adapt the methodology you use. [Natureprotocols.com](https://natureprotocols.com) is fully searchable, providing your protocols and paper with increased utility and visibility. Please submit your protocol to <https://protocolexchange.researchsquare.com/>. After entering your [nature.com](https://www.nature.com) username and password you will need to enter your manuscript number (NG-A56958R4). Further information can be found at <https://www.nature.com/nprot/>.

Sincerely,

Michael Fletcher, PhD
Associate Editor, Nature Genetics

ORCID: 0000-0003-1589-7087